# SimBa: Simplicity Bias for Scaling Up Parameters in Deep Reinforcement Learning

**Hojoon Lee**[1,2*†] **Dongyoon Hwang**[1,3*] **Donghu Kim**[1] **Hyunseung Kim**[1,3] **Jun Jet Tai**[2,4†]
**Kaushik Subramanian**[2] **Peter R.Wurman**[2] **Jaegul Choo**[1] **Peter Stone**[2,5] **Takuma Seno**[2]
[1]KAIST    [2]Sony AI    [3]KRAFTON    [4]Coventry University    [5]UT Austin
{joonleesky, godnpeter}@kaist.ac.kr

## Abstract

Recent advances in CV and NLP have been largely driven by scaling up the number of network parameters, despite traditional theories suggesting that larger networks are prone to overfitting. These large networks avoid overfitting by integrating components that induce a *simplicity bias*, guiding models toward simple and generalizable solutions. However, in deep RL, designing and scaling up networks have been less explored. Motivated by this opportunity, we present *SimBa*, an architecture designed to scale up parameters in deep RL by injecting a simplicity bias. SimBa consists of three components: (i) an observation normalization layer that standardizes inputs with running statistics, (ii) a residual feedforward block to provide a linear pathway from the input to output, and (iii) a layer normalization to control feature magnitudes. By scaling up parameters with SimBa, the sample efficiency of various deep RL algorithms—including off-policy, on-policy, and unsupervised methods—is consistently improved. Moreover, solely by integrating SimBa architecture into SAC, it matches or surpasses state-of-the-art deep RL methods with high computational efficiency across DMC, MyoSuite, and HumanoidBench. These results demonstrate SimBa's broad applicability and effectiveness across diverse RL algorithms and environments.

*Explore codes and videos at https://sonyresearch.github.io/simba*

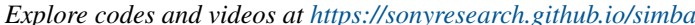

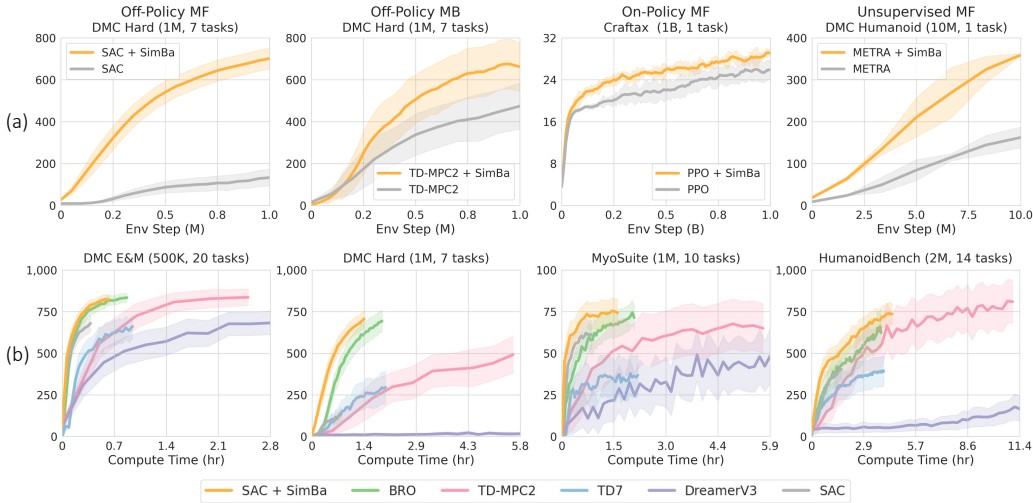

Figure 1: **Benchmark Summary. (a) Sample Efficiency:** SimBa improves sample efficiency across various RL algorithms, including off-policy (SAC, TD-MPC2), on-policy (PPO), and unsupervised RL (METRA). **(b) Compute Efficiency:** When applying SimBa with SAC, it matches or surpasses state-of-the-art off-policy RL methods across 51 continuous control tasks, by only modifying the network architecture and scaling up the number of network parameters.

---

*Equal Contribution

†Work done during internship at Sony AI

# 1 INTRODUCTION

Scaling up neural network sizes has been a key driver of recent advancements in computer vision (CV) (Dehghani et al., 2023) and natural language processing (NLP) (Google et al., 2023; Achiam et al., 2023). By increasing the number of parameters, neural networks gain enhanced expressivity, enabling them to cover diverse functions and discover effective solutions that smaller networks might miss. However, this increased capacity also heightens the risk of overfitting, as larger networks can fit intricate patterns in the training data that do not generalize well to unseen data.

Despite this risk of overfitting, empirical evidence shows that neural networks tend to converge toward simpler functions that generalize effectively (Kaplan et al., 2020; Nakkiran et al., 2021). This phenomenon is attributed to a *simplicity bias* inherent in neural networks, where standard optimization algorithms and architectural components guide highly expressive models toward solutions representing simple, generalizable functions (Shah et al., 2020; Berchenko, 2024). For instance, gradient noise in stochastic gradient descent prevents models from converging on sharp local minima, helping them avoid overfitting (Chizat & Bach, 2020; Gunasekar et al., 2018; Pesme et al., 2021). Architectural components such as ReLU activations (Hermann et al., 2024), layer normalization (Ba et al., 2016), and residual connections (He et al., 2020) are also known to amplify simplicity bias. These components influence the types of functions that neural networks represent at initialization, where networks that represent simpler function at initialization are more likely to converge to simple functions (Valle-Perez et al., 2018; Mingard et al., 2019; Teney et al., 2024).

While scaling up network parameters and leveraging simplicity bias have been successfully applied in CV and NLP, these principles have been underexplored in deep reinforcement learning (RL), where the focus has primarily been on algorithmic advancements (Hessel et al., 2018; Hafner et al., 2023; Fujimoto et al., 2023; Hansen et al., 2023). Motivated by this opportunity, we introduce the *SimBa* network, a novel architecture that explicitly embeds simplicity bias to effectively scale up parameters in deep RL. SimBa comprises of three key components: (i) an observation normalization layer that standardizes inputs by tracking the mean and variance of each dimension, reducing overfitting to high-variance features (Andrychowicz et al., 2020); (ii) a pre-layer normalization residual feedforward block (Xiong et al., 2020), which maintains a direct linear information pathway from input to output and applies non-linearity only when necessary; and (iii) a post-layer normalization before the output layer to stabilize activations, ensuring more reliable policy and value predictions.

To verify whether SimBa amplifies simplicity bias, we compared it against the standard MLP architecture often employed in deep RL. Following Teney et al. (2024), we measured simplicity bias by (i) sampling random inputs from a uniform distribution; (ii) generating network outputs; and (iii) performing Fourier decomposition on these outputs. A smaller sum of the Fourier coefficients indicates that the neural network represents a low-frequency function, signifying greater simplicity. We define the simplicity score as the inverse of this sum, meaning a higher score corresponds to a stronger simplicity bias. As illustrated in Figure 2.(a), our analysis revealed that SimBa has a higher simplicity score than the MLP (Further details are provided in Section 2).

To evaluate the benefit of leveraging simplicity bias on network scaling, we compared the performance of Soft Actor-Critic (Haarnoja et al., 2018, SAC) using both MLP and our SimBa architecture across 3 humanoid tasks from DMC benchmark (Tassa et al., 2018). We increased the width of both actor and critic networks, scaling from 0.1 to 17 million parameters. As shown in Figure 2.(b), SAC with MLP experiences performance degradation as the number of parameters increases. In contrast, SAC with the SimBa network consistently improves its performance as the number of parameter increases, highlighting the value of embedding simplicity bias when scaling deep RL networks.

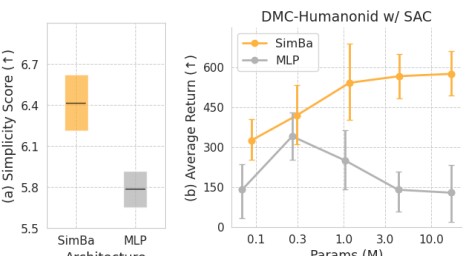

Figure 2: (a) SimBa exhibits higher simplicity bias than MLP. (b) SAC with SimBa improves its performance with increased parameters, whereas SAC with MLP degrades it. Each standard deviation is 95% CI.

To further evaluate SimBa's versatility, we applied it to various RL algorithms by only changing the network architecture and scaling up parameters. The algorithms included off-policy (SAC (Haarnoja et al., 2018), TD-MPC2 (Hansen et al., 2023)), on-policy model-free (PPO (Schulman et al., 2017)),

and unsupervised RL (METRA (Park et al., 2023)). As illustrated in Figure 1.(a), SimBa consistently enhances the sample efficiency of these algorithms. Furthermore, as shown in Figure 1.(b), when SimBa is integrated into SAC, it matches or surpasses state-of-the-art off-policy methods across 51 tasks in DMC, MyoSuite (Caggiano et al., 2022), and HumanoidBench (Sferrazza et al., 2024). Despite the increased number of parameters, SAC with SimBa remains computationally efficient because it does not employ any computationally intensive components such as self-supervised objectives (Fujimoto et al., 2023), planning (Hansen et al., 2023), or replay ratio scaling (Nauman et al., 2024), which state-of-the-art methods rely on to achieve high performance.

## 2 PRELIMINARY

Simplicity bias refers to the tendency of neural networks to prioritize learning simpler patterns over capturing intricate details (Shah et al., 2020; Berchenko, 2024). In this section, we introduce metrics to quantify simplicity bias; in-depth definitions are provided in Appendix A.

### 2.1 MEASURING FUNCTION COMPLEXITY

We begin by defining a **complexity measure** $c : \mathcal{F} \to [0, \infty)$ to quantify the complexity of functions in $\mathcal{F} = \{f | f : \mathcal{X} \to \mathcal{Y}\}$ where $\mathcal{X} \subseteq \mathbb{R}^n$ denote the input space and $\mathcal{Y} \subseteq \mathbb{R}^m$ the output space.

Traditional complexity measures, such as the Vapnik–Chervonenkis dimension (Blumer et al., 1989) and Rademacher complexity (Bartlett & Mendelson, 2002), are well-established but often intractable for deep neural networks. Therefore, we follow Teney et al. (2024) and adopt Fourier analysis as our primary complexity measure. Given $f(x) := (2\pi)^{d/2} \int \tilde{f}(k) \, e^{ik \cdot x} dk$ where $\tilde{f}(k) := \int f(x) \, e^{-ik \cdot x} dx$ is the Fourier transform, we perform a discrete Fourier transform by uniformly discretizing the frequency domain, $k \in \{0, 1, \ldots, K\}$. The value of $K$ is chosen by the Nyquist-Shannon limit (Shannon, 1949) to ensure accurate function representation. Our complexity measure $c(f)$ is then computed as the frequency-weighted average of the Fourier coefficients:

$$c(f) \;=\; \Sigma_{k=0}^{K} \tilde{f}(k) \cdot k \;/\; \Sigma_{k=0}^{K} \tilde{f}(k). \tag{1}$$

Intuitively, larger $c(f)$ indicates higher complexity due to a dominance of high-frequency components such as rapid amplitude changes or intricate details. Conversely, lower $c(f)$ implies a larger contribution from low-frequency components, indicating a lower complexity function.

### 2.2 MEASURING SIMPLICITY BIAS

In theory, simplicity bias can be measured by evaluating the complexity of the function to which the network converges after training. However, directly comparing simplicity bias across different architectures after convergence is challenging due to the randomness of the non-stationary optimization process, especially in RL, where the data distribution changes continuously.

Empirical studies suggest that the initial complexity of a network strongly correlates with the complexity of the functions it converges to during training (Valle-Perez et al., 2018; De Palma et al., 2019; Mingard et al., 2019; Teney et al., 2024). Therefore, for a given network architecture $f$ with an initial parameter distribution $\Theta_0$, we define the **simplicity bias score** $s(f) : \mathcal{F} \to (0, \infty)$ as:

$$s(f) \approx \mathbb{E}_{\theta \sim \Theta_0} \left[ \frac{1}{c(f_\theta)} \right] \tag{2}$$

where $f_\theta$ denotes the network architecture $f$ parameterized by $\theta$.

This measure indicates that networks with lower complexity at initialization are more likely to exhibit a higher simplicity bias, thereby converging to simpler functions during training.

## 3 RELATED WORK

### 3.1 SIMPLICITY BIAS

Initially, simplicity bias was mainly attributed to the implicit regularization effects of the stochastic gradient descent (SGD) optimizer (Soudry et al., 2018; Gunasekar et al., 2018; Chizat & Bach, 2020;

Figure 3: **SimBa architecture.** The network integrates Running Statistics Normalization (RSNorm), Residual Feedforward Blocks, and Post-Layer Normalization to embed simplicity bias into deep RL.

Pesme et al., 2021). During training, SGD introduces noise which prevents the model from converging to sharp minima, guiding it toward flatter regions of the loss landscape (Wu et al., 2022). Such flatter minima are associated with functions of lower complexity, thereby improving generalization.

However, recent studies suggest that simplicity bias is also inherent in the network architecture itself (Valle-Perez et al., 2018; Mingard et al., 2019). Architectural components such as normalization layers, ReLU activations (Hermann et al., 2024), and residual connections (He et al., 2020) promote simplicity bias by encouraging smoother, less complex functions. Fourier analysis has shown that these components help models prioritize learning low-frequency patterns, guiding optimization toward flatter regions that generalize better (Teney et al., 2024). Consequently, architectural design plays a crucial role in favoring simpler solutions, enabling the use of overparameterized networks.

### 3.2 DEEP REINFORCEMENT LEARNING

For years, deep RL has largely focused on algorithmic improvements to enhance sample efficiency and generalization. Techniques like Double Q-learning (Van Hasselt et al., 2016; Fujimoto et al., 2018), and Distributional RL (Dabney et al., 2018) have improved the stability of value estimation by reducing overestimation bias. Regularization strategies—including periodic reinitialization (Nikishin et al., 2022; Lee et al., 2024), layer normalization (Lee et al., 2023; Gallici et al., 2024), batch normalization (Bhatt et al., 2024), and spectral normalization (Gogianu et al., 2021)—have been employed to prevent overfitting and enhance generalization. Incorporating self-supervised objectives with model-based learning (Fujimoto et al., 2023; Hafner et al., 2023; Hansen et al., 2023) has further improved representation learning and sample efficiency.

Despite these advances, scaling up network architectures in deep RL remains underexplored, especially by leveraging the simplicity bias principle. Several recent studies have attempted to scale up network sizes—through ensembling (Chen et al., 2021; Obando-Ceron et al., 2024), widening (Schwarzer et al., 2023), residual connections (Espeholt et al., 2018) and deepening networks (Bjorck et al., 2021; Nauman et al., 2024). However, they often rely on computationally intensive layers such as spectral normalization (Bjorck et al., 2021) or require sophisticated training protocols (Nauman et al., 2024), limiting their applicability.

In this work, we aim to design an architecture that amplifies simplicity bias, enabling us to effectively scale up parameters in RL, independent of using any other sophisticated training protocol.

## 4 SIMBA

This section introduces the SimBa network, an architecture designed to embed simplicity bias into deep RL. The architecture is composed of Running Statistics Normalization, Residual Feedforward Blocks, and Post-Layer Normalization. By amplifying the simplicity bias, SimBa allows the model to avoid overfitting for highly overparameterized configurations.

**Running Statistics Normalization (RSNorm).** First, RSNorm standardizes input observations by tracking the running mean and variance of each input dimension during training, preventing features with disproportionately large values from dominating the learning process.

Given an input observation $\mathbf{o}_t \in \mathbb{R}^{d_o}$ at timestep $t$, we update the running observation mean $\mu_t \in \mathbb{R}^{d_o}$ and variance $\sigma_t^2 \in \mathbb{R}^{d_o}$ as follows:

$$\mu_t = \mu_{t-1} + \frac{1}{t}\delta_t, \quad \sigma_t^2 = \frac{t-1}{t}(\sigma_{t-1}^2 + \frac{1}{t}\delta_t^2) \tag{3}$$

where $\delta_t = \mathbf{o}_t - \mu_{t-1}$ and $d_o$ denotes the dimension of the observation.

Once $\mu_t$ and $\sigma_t^2$ are computed, each input observation $\mathbf{o}_t$ is normalized as:

$$\bar{\mathbf{o}}_t = \text{RSNorm}(\mathbf{o}_t) = \frac{\mathbf{o}_t - \mu_t}{\sqrt{\sigma_t^2 + \epsilon}} \tag{4}$$

where $\bar{\mathbf{o}}_t \in \mathbb{R}^{d_o}$ is the normalized output, and $\epsilon$ is a small constant for numerical stability.

While alternative observation normalization methods exist, RSNorm consistently demonstrates superior performance. A comprehensive comparison is provided in Section 7.1.

**Residual Feedforward Block.** The normalized observation $\bar{\mathbf{o}}_i$ is first embedded into a $d_h$-dimensional vector using a linear layer:

$$\mathbf{x}_t^l = \text{Linear}(\bar{\mathbf{o}}_t). \tag{5}$$

At each block, $l \in \{1, ..., L\}$, the input $\mathbf{x}_t^l$ passes through a pre-layer normalization residual feedforward block, introducing simplicity bias by allowing a direct linear pathway from the input to the output. This direct linear pathway enables the network to pass on the input unchanged throughout the entire network unless non-linear transformations are necessary. Each block is defined as:

$$\mathbf{x}_t^{l+1} = \mathbf{x}_t^l + \text{MLP}(\text{LayerNorm}(\mathbf{x}_t^l)). \tag{6}$$

Following Vaswani (2017), the MLP is structured with an inverted bottleneck, where the hidden dimension is expanded to $4 \cdot d_h$ and a ReLU activation is applied between the two linear layers.

**Post-Layer Normalization.** To ensure that activations remain on a consistent scale before predicting the policy or value function, we apply layer normalization after the final residual block:

$$\mathbf{z}_t = \text{LayerNorm}(\mathbf{x}_t^L). \tag{7}$$

The normalized output $\mathbf{z}_t$ is then processed through a linear layer, generating predictions for the actor's policy or the critic's value function.

## 5 ANALYZING SIMBA

In this section, we analyze whether each component of SimBa amplifies simplicity bias and allows scaling up parameters in deep RL. We conducted experiments on challenging environments in DMC involving Humanoid and Dog, collectively referred to as DMC-Hard. Throughout this section, we used Soft Actor Critic (Haarnoja et al., 2018, SAC) as the base algorithm.

### 5.1 ARCHITECTURAL COMPONENTS

To quantify simplicity bias in SimBa, we use Fourier analysis as described in Section 2. For each network $f$, we estimate the simplicity bias score $s(f)$ by averaging over 100 random initializations $\theta \sim \Theta_0$. Following Teney et al. (2024), the input space $\mathcal{X} = [-100, 100]^2 \subset \mathbb{R}^2$ is divided into a grid of 90,000 points, and the network outputs a scalar value for each input which are represented as a grayscale image. By applying the discrete Fourier transform to these outputs, we compute the simplicity score $s(f)$ defined in Equation 2 (see Appendix B for further details).

Figure 4.(a) shows that SimBa's key components—such as residual connections and layer normalization—increase the simplicity bias score $s(f)$, biasing the architecture toward simpler functional representations at initialization. When combined, these components induce a stronger preference for low-complexity functions (i.e., high simplicity score) than when used individually.

Figure 4.(b) reveals a clear relationship between simplicity bias and performance: architectures with higher simplicity bias scores lead to enhanced performance. Specifically, compared to the MLP, adding residual connections increases the average return by 50 points, adding layer normalization adds 150 points, and combining all components results in a substantial improvement of 550 points.

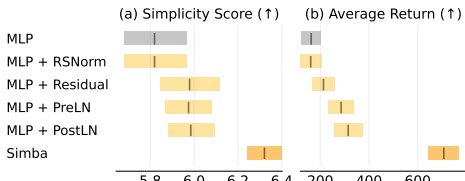

Figure 4: **Component Analysis.** **(a)** Simplicity bias scores estimated via Fourier analysis. Mean and 95% CI are computed over 100 random initializations. **(b)** Average return in DMC-Hard for 1M steps. Mean and 95% CI over 10 seeds, using SAC. Stronger simplicity bias correlates with higher returns for overparameterized networks.

## 5.2 COMPARISON WITH OTHER ARCHITECTURES

To assess SimBa's scalability, we compared it to BroNet (Nauman et al., 2024), SpectralNet (Bjorck et al., 2021), and MLP. Detailed descriptions of each architecture are provided in Appendix F.

The key differences between SimBa and other architectures lie in the placement of components that promote simplicity bias. SimBa maintains a direct linear residual pathway from the input to the output, applying non-linearity exclusively through residual connections. In contrast, other architectures introduce non-linearities within the input-to-output pathway, increasing functional complexity. Additionally, SimBa employs post-layer normalization to ensure consistent activation scales across all hidden dimensions, reducing variance in policy and value predictions. Conversely, other architectures omit normalization before the output layer, which can lead to high variance in their predictions.

To ensure that performance differences were attributable to architectural design choices rather than varying input normalization, we uniformly applied RSNorm as the input normalization across all models. Our investigation focused on scaling up the critic network's parameters by increasing the hidden dimension, as scaling up the actor network showed limited benefits (see Section 7.2).

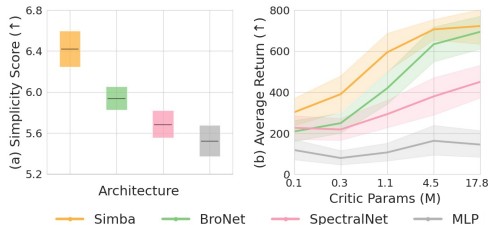

As illustrated in Figure 5.(a), SimBa achieves the highest simplicity bias score compared to the experimented architectures, demonstrating its strong preference for simpler solutions. In Figure 5.(b), while the MLP failed to scale with increasing network size, BroNet, SpectralNet, and SimBa all showed performance improvements.

Importantly, the scalability of each architecture is correlated with its simplicity bias score, with a higher simplicity bias leading to better scalability. SimBa demonstrated the best scalability, supporting the hypothesis that simplicity bias plays a key role in scaling deep RL. Further analysis on this correlation is provided in Appendix D.

Figure 5: **Architecture Comparison**. **(a)** SimBa consistently exhibits a higher simplicity bias score. **(b)** SimBa demonstrates strong scaling performance in terms of average return for DMC-Hard compared to the other architectures. The results are from 5 random seeds.

## 6 EXPERIMENTS

This section evaluates SimBa's applicability across various deep RL algorithms and environments. For each baseline, we either use the authors' reported results or run experiments using their recommended hyperparameters. Detailed descriptions of the environments used in our evaluations are provided in Appendix H. In addition, we also include learning curves and final performance for each task in Appendix J and K, along with hyperparameters used in our experiments in Appendix I.

### 6.1 OFF-POLICY RL

**Experimental Setup.** We evaluate the algorithms on 51 tasks across three benchmarks: DMC (Tassa et al., 2018), MyoSuite (Caggiano et al., 2022), and HumanoidBench (Sferrazza et al., 2024). The

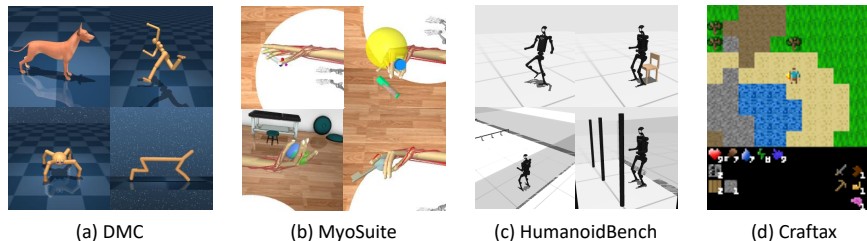

(a) DMC   (b) MyoSuite   (c) HumanoidBench   (d) Craftax

Figure 6: **Environment Visualizations.** SimBa is evaluated across four diverse benchmark environments: DMC, MyoSuite, and HumanoidBench, which feature complex locomotion and manipulation tasks, and Craftax, which introduces open-ended tasks with varying complexity.

Figure 8: **Off-policy RL Benchmark.** Average episode return for DMC and HumanoidBench and average success rate for MyoSuite across 51 continuous control tasks. SimBa (with SAC) achieves high computational efficiency by only changing the network architecture.

DMC tasks are further categorized into DMC-Easy&Medium and DMC-Hard based on complexity. We vary the number of training steps per benchmark according to the complexity of the environment: 500K for DMC-Easy&Medium, 1M for DMC-Hard and MyoSuite, and 2M for HumanoidBench.

**Baselines.** We compare SimBa against state-of-the-art off-policy RL algorithms. (i) SAC (Haarnoja et al., 2018), an actor-critic algorithm based on maximum entropy RL; (ii) DDPG (Lillicrap, 2015), a deterministic policy gradient algorithm with deep neural networks; (iii) TD7 (Fujimoto et al., 2023), an enhanced version of TD3 incorporating state-action representation learning; (iv) BRO (Nauman et al., 2024), which scales the critic network of SAC while integrating distributional Q-learning, optimistic exploration, and periodic resets. For a fair comparison, we use BRO-Fast, which is the most computationally efficient version; (v) TD-MPC2 (Hansen et al., 2023), which combines trajectory planning with long-return estimates using a learned world model; and (vi) DreamerV3 (Hafner et al., 2023), which learns a generative world model and optimizes a policy via simulated rollouts.

**Integrating SimBa into Off-Policy RL.** The SimBa architecture is algorithm-agnostic and can be applied to any deep RL method. To demonstrate its applicability, we integrate SimBa into two model-free (SAC, DDPG) and one model-based algorithm (TD-MPC2), evaluating their performance on the DMC-Hard benchmark. For SAC and DDPG, we replace the standard MLP-based actor-critic networks with SimBa. For TD-MPC2, we substitute the shared encoder from MLP to SimBa while matching the number of parameters as the original implementation.

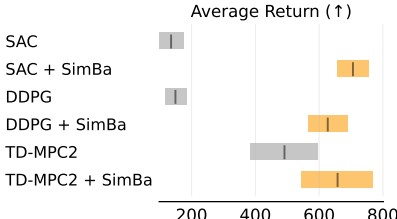

Figure 7: **Off-Policy RL with SimBa.** Replacing MLP with SimBa leads to substantial performance improvements across various off-policy RL methods. Mean and 95% CI are averaged over 10 seeds for SAC and DDPG, and 3 seeds for TD-MPC2 in DMC-Hard.

Figure 7 shows consistent benefits across all algorithms: integrating SimBa increased the average return by 570, 480, and, 170 points for SAC, DDPG, and TD-MPC2 respectively. These results demonstrate that scaling up parameters with a strong simplicity bias significantly enhances performance in deep RL.

**Comparisons with State-of-the-Art Methods.** Here, we compare SAC + SimBa against state-of-the-art off-policy deep RL algorithms, which demonstrated the most promising results in previous experiments. Throughout this section, we refer to SAC + SimBa as SimBa.

Figure 8 presents the results, where the x-axis represents computation time using an RTX 3070 GPU, and the y-axis denotes performance. Points in the upper-left corner (↖) indicate higher compute efficiency. Here, SimBa consistently outperforms most baselines across various environments while requiring less computational time. The only exception is TD-MPC2 on HumanoidBench; however, TD-MPC2 requires 2.5 times the computational resources of SimBa to achieve better performance, making SimBa the most efficient choice overall.

The key takeaway is that SimBa achieves these remarkable results without the bells and whistles often found in state-of-the-art deep RL algorithms. It is easy to implement, which only requires modifying the network architecture into SimBa without additional changes to the loss functions (Schwarzer et al., 2020; Hafner et al., 2023) or using domain-specific regularizers (Fujimoto et al., 2023; Nauman et al., 2024). Its effectiveness stems from an architectural design that leverages simplicity bias and scaling up the number of parameters, thereby maximizing performance.

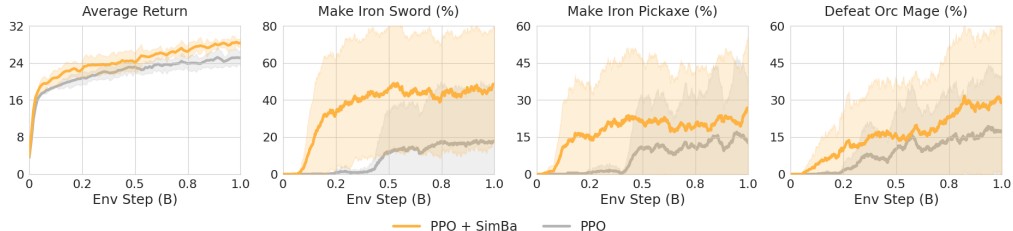

Figure 9: **Impact of Input Dimension.** Average episode return for DMC tasks plotted against increasing state dimensions. Results show that the benefits of using SimBa increase with higher input dimensions, effectively alleviating the curse of dimensionality.

Figure 10: **On-policy RL with SimBa.** Average return of achievements and task success rate for three different tasks comparing PPO + SimBa and PPO on Craftax. Integrating SimBa enables effective learning of complex behaviors.

**Impact of Input Dimension.** To further identify in which cases SimBa offers significant benefits, we analyze its performance across DMC environments with varying input dimensions. As shown in Figure 9, the advantage of SimBa becomes more pronounced as input dimensionality increases. We hypothesize that higher-dimensional inputs exacerbate the curse of dimensionality, and the simplicity bias introduced by SimBa effectively mitigates overfitting in these high-dimensional settings.

## 6.2 ON-POLICY RL

**Experimental Setup.** We conduct our on-policy RL experiments in Craftax (Matthews et al., 2024), an open-ended environment inspired by Crafter (Hafner, 2022) and NetHack (Küttler et al., 2020). Craftax poses a unique challenge with its open-ended structure and compositional tasks. Following Matthews et al. (2024), we use Proximal Policy Optimization (Schulman et al., 2017, PPO) as the baseline algorithm. When integrating SimBa with PPO, we replace the standard MLP-based actor-critic networks with SimBa and train both PPO and PPO + SimBa on 1024 parallel environments for a total of 1 billion environment steps.

**Results.** As illustrated in Figure 10, integrating SimBa into PPO significantly enhances performance across multiple complex tasks. With SimBa, the agent learns to craft iron swords and pickaxes more rapidly, enabling the early acquisition of advanced tools. Notably, by using these advanced tools, the SimBa-based agent successfully defeats challenging adversaries like the Orc Mage using significantly fewer time steps than an MLP-based agent. These improvements arise solely from the architectural change, demonstrating the effectiveness of the SimBa approach for on-policy RL.

## 6.3 UNSUPERVISED RL

**Experimental Setup.** In our unsupervised RL study, we incorporate SimBa for online skill discovery, aiming to identify diverse behaviors without relying on task-specific rewards. We focus our experiments primarily on METRA (Park et al., 2023), which serves as the state-of-the-art algorithm in this domain. We evaluate METRA and METRA with SimBa on the Humanoid task from DMC, running 10M environment steps. For a quantitative comparison, we adopt state coverage as our main metric. Coverage is measured by discretizing the $x$ and $y$ axes into a grid and counting the number of grid cells covered by the learned behaviors at each evaluation epoch, following prior literature (Park et al., 2023; Kim et al., 2024a;b).

**Results.** As illustrated in Figure 1.(a) and 11, integrating SimBa into METRA significantly enhances state coverage on the Humanoid task. The high-dimensional input space makes it challenging for METRA to learn diverse skills. By injecting simplicity bias to manage high input dimensions and scaling up parameters to facilitate effective diverse skill acquisition, SimBa effectively leads to improved exploration and broader state coverage.

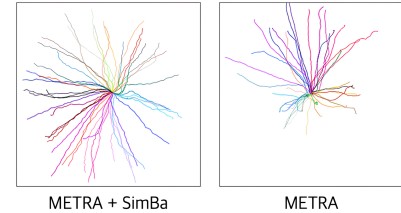

METRA + SimBa          METRA

Figure 11: **URL with SimBa**. Integrating SimBa to METRA enhances state coverage.

## 7 ABLATIONS

We conducted ablations on DMC-Hard with SAC + SimBa, running 5 seeds for each experiment.

### 7.1 OBSERVATION NORMALIZATION

A key factor in SimBa's success is using RSNorm for observation normalization. To validate its effectiveness, we compare 6 alternative methods: (i) LayerNorm (Ba et al., 2016); (ii) RMSNorm (Zhang & Sennrich, 2019); (iii) BatchNorm (Ioffe & Szegedy, 2015); (iv) Env Wrapper RSNorm, which tracks running statistics for each dimension, normalizes observations upon receiving from the environment, and stores them in the replay buffer; (v) Initial N Steps, which fixes the statistics derived from the initially collected transition samples, where we used $N = 5,000$; and (vi) Oracle Statistics, which rely on pre-computed statistics from previously collected expert data.

As illustrated in Figure 12, layer normalization and batch normalization offer little to no performance gains. While the env-wrapper RSNorm is somewhat effective, it falls short of RSNorm's performance. Although widely used in deep RL frameworks (Dhariwal et al., 2017; Hoffman et al., 2020; Raffin et al., 2021), the env-wrapper introduces inconsistencies in off-policy settings by normalizing samples with different statistics based on their collection time. This causes identical observations to be stored with varying values in the replay buffer, reducing learning consistency. Fixed initial statistics also show slightly worse performance than RSNorm, potentially due to their inability to adapt to the evolving dynamics during training. Overall, only RSNorm matched the performance of Oracle statistics, making it the most practical observation normalization choice for deep RL.

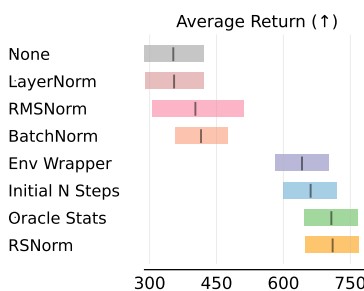

Figure 12: **Obs Normalization**. RSNorm consistently outperforms alternative normalization methods. Mean and 95% CI over 5 seeds.

### 7.2 SCALING THE NUMBER OF PARAMETERS

Here, we investigate scaling parameters in the actor-critic architecture with SimBa by focusing on two aspects: (i) scaling the actor versus the critic network, and (ii) scaling network width versus depth. For width scaling, the actor and critic depths are set to 1 and 2 blocks, respectively. For depth scaling, the actor and critic widths are fixed at 128 and 512, respectively, following our default setup.

As illustrated in Figure 13, scaling up the width or depth of the critic (➡) generally improves performance, while scaling up the actor's width or depth (⬆) tends to reduce performance. This contrast suggests that the target complexity of the actor may be lower than that of the critic, where scaling up the actor's parameters might be ineffective. These findings align with previous study (Nauman et al., 2024), which highlights the benefits of scaling up the critic while showing limited advantages in scaling up the actor.

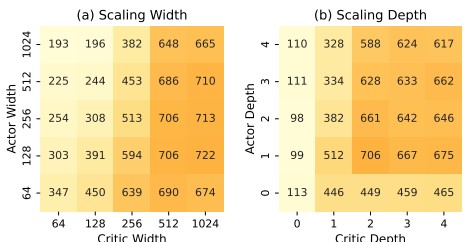

Figure 13: **Scaling Actor and Critic Network.** Performance of SAC with SimBa by varying width and depth for the actor and critic network.

Furthermore, for the critic network, scaling up the width is generally more effective than scaling up the depth. While both approaches can enhance the network's expressivity, scaling up the depth can decrease the simplicity bias as it adds more non-linear components within the network. Based on our findings, we recommend width scaling as the default strategy.

## 7.3 SCALING REPLAY RATIO

According to scaling laws (Kaplan et al., 2020), performance can be enhanced not only by scaling up the number of parameters but also by scaling up the computation time, which can be done by scaling up the number of gradient updates per collected sample (i.e., replay ratio) in deep RL.

However, in deep RL, scaling up the replay ratio has been shown to decrease performance due to the risk of overfitting the network to the initially collected samples (Nikishin et al., 2022; D'Oro et al., 2022; Lee et al., 2023). To address this issue, recent research has proposed periodically reinitializing the network to prevent overfitting, demonstrating that performance can be scaled with respect to the replay ratio. In this section, we aim to assess whether the simplicity bias induced by SimBa can mitigate overfitting under increased computation time (i.e., higher replay ratios).

To evaluate this, we trained SAC with SimBa using 2, 4, 8, and 16 replay ratios, both with and without periodic resets. For SAC with resets, we reinitialized the entire network and optimizer every 500,000 gradient steps. We also compared our results with BRO (Nauman et al., 2024), which incorporates resets.

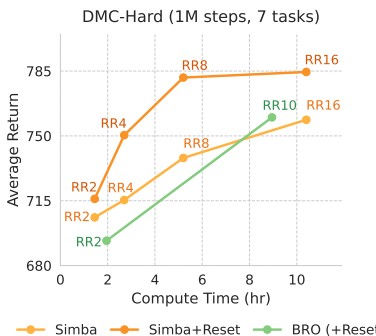

Surprisingly, as illustrated in Figure 14, SimBa's performance consistently improves as the replay ratio increases, even without periodic resets. We have excluded results for BRO without resets, as it fails to learn meaningful behavior for all replay ratios (achieves lower than 300 points). Notably, when resets are included for SimBa, the performance gains become even more pronounced, with a replay ratio of 8 outperforming the most computationally intensive BRO algorithm (RR10).

Figure 14: **Scaling Replay Ratio.** Performance of SimBa with and without periodic resets for various replay ratios.

## 8 LESSONS AND OPPORTUNITIES

**Lessons.** Deep reinforcement learning has historically struggled with overfitting, requiring complex training protocols and tricks to mitigate these issues (Hessel et al., 2018; Fujimoto et al., 2023). These complexities can hinder practitioners with limited resources. In this paper, we improved performance solely by modifying the network architecture while keeping the underlying algorithms unchanged, simplifying SimBa's implementation and making it easy to adopt. By incorporating simplicity bias through architectural design and scaling up parameters, our network converges to simpler, generalizable functions, matching or surpassing state-of-the-art methods. This aligns with Richard Sutton's Bitter Lesson (Sutton, 2019): while task-specific designs may offer immediate gains, scalable approaches provide more sustainable long-term benefits.

**Opportunities.** While our exploration has focused on network architecture, optimization techniques such as dropout (Hiraoka et al., 2021), data augmentation (Kostrikov et al., 2020), and advanced optimization algorithms (Foret et al., 2020) are also crucial for promoting convergence to simpler functions. Extending our insights to vision-based RL presents a promising direction; specifically, designing convolutional architectures guided by simplicity bias principles could substantially enhance learning efficiency in visual tasks. Furthermore, integrating SimBa with multi-task RL may improve performance across diverse tasks by facilitating the learning of shared, simple, and generalizable representations across tasks. Although our current work centers on model-free RL algorithms, model-based approaches have also demonstrated considerable success (Schwarzer et al., 2020; Hansen et al., 2023; Hafner et al., 2023). Our preliminary investigations applying SimBa to model-based RL, specifically TD-MPC2 (Hansen et al., 2023), have yielded promising results that warrant further exploration. We encourage the research community to pursue these directions to advance deep RL architectures and facilitate their successful application in real-world scenarios.

REPRODUCIBILITY

To ensure the reproducibility of our experiments, we provide the complete source code to run SAC with SimBa. Moreover, we have included a Dockerfile to simplify the testing and replication of our results. The raw scores of each experiment including all baselines are also included. The code is available at https://github.com/SonyResearch/simba.

ACKNOWLEDGEMENT

This work was supported by the Institute for Information & communications Technology Planning & Evaluation(IITP) grant funded by the Korea government(MSIT) (RS-2019-II190075, Artificial Intelligence Graduate School Program(KAIST)) and the National Research Foundation of Korea(NRF) grant funded by the Korea government(MSIT) (No. RS-2025-00555621). We would also like to express our gratitude to Craig Sherstan, Bram Grooten, Hanseul Cho, Kyungmin Lee, and Hawon Jeong for their invaluable discussions and insightful feedback, which have significantly contributed to this work.

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

# APPENDIX

# A    DEFINITION OF SIMPLICITY BIAS

This section provides more in-depth definition of simplicity bias for clarity.

## A.1    COMPLEXITY MEASURE

Let $\mathcal{X} \subseteq \mathbb{R}^n$ denote the input space and $\mathcal{Y} \subseteq \mathbb{R}^m$ the output space. Consider the function space $\mathcal{F} = \{f | f : \mathcal{X} \to \mathcal{Y}\}$. Given a training dataset $D = \{(x_i, y_i)\}_{i=1}^N$ and a tolerance $\epsilon > 0$, define the set of functions $\mathcal{I} \subseteq \mathcal{F}$ achieving a training loss below $\epsilon$:

$$\mathcal{I} = \left\{ f \in \mathcal{F} \; \middle| \; \mathcal{L}_{\text{train}}(f) = \frac{1}{N} \sum_{i=1}^N \ell(f(x_i), y_i) < \epsilon \right\}, \tag{8}$$

where $\ell : \mathcal{Y} \times \mathcal{Y} \to [0, \infty)$ is a loss function such as mean squared error or cross-entropy.

To quantify the complexity of functions within $\mathcal{F}$, we adopt a Fourier-based complexity measure as described in Section 2. Specifically, for a function $f$ with a Fourier series representation:

$$f(x) = (2\pi)^{d/2} \int \tilde{f}(k) \, e^{ik \cdot x} \, dk, \tag{9}$$

where $\tilde{f}(k) = \int f(x) \, e^{-ik \cdot x} \, dx$ is the Fourier transform of $f$.

Given a Fourier series representation $\tilde{f}$, we perform a discrete Fourier transform by uniformly discretizing the frequency domain. Specifically, we consider frequencies $k \in \{0, 1, \dots, K\}$, where $K$ is selected based on the Nyquist-Shannon sampling theorem (Shannon, 1949) to ensure an accurate representation of $f$. This discretization transforms the continuous frequency domain into a finite set of frequencies suitable for computational purposes.

Then, the **complexity measure** $c : \mathcal{F} \to [0, \infty)$ is defined as the weighted average of the Fourier coefficients from discretization:

$$c(f) = \frac{\sum_{k=0}^K \tilde{f}(k) \cdot k}{\sum_{k=0}^K \tilde{f}(k)}. \tag{10}$$

Equation 10 captures the intuition that higher complexity arises from the dominance of high-frequency components, which correspond to rapid amplitude changes or intricate details in the function $f$. Conversely, a lower complexity measure indicates a greater contribution from low-frequency components, reflecting simpler, smoother functions.

## A.2    SIMPLICITY BIAS SCORE

Let $f$ be a neural network, and let alg $\in \mathcal{A}$ denote the optimization algorithm used for training, such as stochastic gradient descent. We quantify the **simplicity bias score** $s : \mathcal{F} \times \mathcal{A} \to (0, \infty)$ as:

$$s(f, \text{alg}) = \mathbb{E}_{f^* \sim P_{f, \text{alg}}} \left[ \frac{1}{c(f^*)} \right] \tag{11}$$

where $P_{f, \text{alg}}$ denotes the distribution over $\mathcal{I}$ induced by training $f$ with alg. A higher simplicity bias score indicates a stronger tendency to converge toward functions with lower complexity.

Then, from equation 11, we can compute the simplicity bias score with respect to the architecture by marginalizing over the distribution of the algorithm as:

$$s(f) = \mathbb{E}_{\text{alg} \sim \mathbf{A}} \left[ \mathbb{E}_{f^* \sim P_{f, \text{alg}}} \left[ \frac{1}{c(f^*)} \right] \right]. \tag{12}$$

Directly measuring this bias by analyzing the complexity of the converged function is challenging due to the non-stationary nature of training dynamics, especially in reinforcement learning where data distributions evolve over time. Empirical studies (Valle-Perez et al., 2018; De Palma et al., 2019; Mingard et al., 2019; Teney et al., 2024) suggest that the initial complexity of a network is strongly correlated with the complexity of the functions it learns after training.

Therefore, as a practical proxy for simplicity bias, we define the approximate of the simplicity bias score $s(f)$ for a network architecture $f$ with an initial parameter distribution $\Theta_0$ as:

$$s(f) \approx \mathbb{E}_{\theta \sim \Theta_0} \left[ \frac{1}{c(f_\theta)} \right]. \tag{13}$$

where $f_\theta$ denotes the network $f$ parameterized by $\theta$.

A higher simplicity bias score indicates that, on average, the network initialized from $\Theta_0$ represents simpler functions prior to training, suggesting a predisposition to converge to simpler solutions during optimization. This measure aligns with the notion that networks with lower initial complexity are more likely to exhibit a higher simplicity bias.

## B   MEASURING SIMPLICITY BIAS

To quantify the simplicity bias defined in Equation 13, we adopt a methodology inspired by the Neural Redshift framework (Teney et al., 2024). We utilize the original codebase provided by the authors[1] to assess the complexity of random neural networks. This approach evaluates the inherent complexity of the architecture before any optimization, thereby isolating the architectural effects from those introduced by training algorithms such as stochastic gradient descent.

Following the methodology outlined in Teney et al. (2024), we perform the following steps to measure the simplicity bias score of a neural network $f_\theta(\cdot)$:

**Sampling Initialization.** For each network architecture $f$, we generate $N$ random initializations $\theta \sim \Theta_0$. This ensemble of initial parameters captures the variability in complexity introduced by different random seeds. In this work, we used $N = 100$.

**Sampling Input.** We define the input space $\mathcal{X} = [-100, 100]^2 \subset \mathbb{R}^2$ and uniformly divide it into a grid of 90,000 points, achieving 300 divisions along each dimension. This dense sampling ensures comprehensive coverage of the input domain.

**Evaluating Function.** For each sampled input point $x \in \mathcal{X}$, we compute the corresponding output $f_\theta(x)$. The collection of these output values forms a 2-D grid of scalars, effectively representing the network's function as a grayscale image.

**Discrete Fourier Transform.** We apply a discrete Fourier transform (DFT) to the grayscale image obtained from the function evaluations. This transformation decomposes $f_\theta(\cdot)$ into a sum of basis functions of varying frequencies.

**Measuring Function Complexity.** Utilizing the Fourier coefficients obtained from the DFT, we compute the complexity measure $c(f_\theta)$ as defined in Equation 10. Specifically, we calculate the frequency-weighted average of the Fourier coefficients:

$$c(f_\theta) = \frac{\sum_{k=0}^{K} \tilde{f}_\theta(k) \cdot k}{\sum_{k=0}^{K} \tilde{f}_\theta(k)}, \tag{14}$$

where $\tilde{f}_\theta(k)$ denotes the Fourier coefficient at frequency $k$.

**Estimating Simplicity Bias Score.** For each network $f$, we estimate the simplicity bias score $s(f)$ by averaging the inverse of the complexity measures over all $N$ random initializations:

$$s(f) \approx \frac{1}{N} \sum_{i=1}^{N} \frac{1}{c(f_{\theta_i})}, \tag{15}$$

where $\theta_i$ represents the $i$-th random initialization.

---

[1] https://github.com/ArmandNM/neural-redshift

## C  ADDITIONAL ABLATIONS

### C.1  ARCHITECTURAL ABLATION

**Experimental Setup.** Following the architectural component analysis in Section 5.1, we further conducted ablation studies by removing each key component from SimBa to assess its impact on the simplicity score and average return. We performed these experiments on DMC-Hard using the same training setup as in Section 5.1, training for 1M environment steps.

**Results.** As shown in Figure 15.(a), we observe that, except for RSNorm, removing key components of SimBa—such as residual connections, pre-layer normalization, or post-layer normalization—leads to a reduction in the simplicity bias score. This indicates that the functional representation at initialization becomes biased toward more complex functions when these components are absent. This finding further emphasizes the critical role of each component in promoting simplicity bias within SimBa's design. Additionally, Figure 15.(b) demonstrates that removing any key component significantly impacts SimBa's performance, underscoring the necessity of each architectural component.

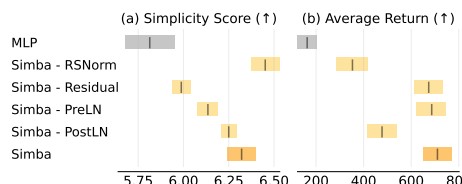

Figure 15: **Additional Component Analysis.** **(a)** Simplicity bias scores estimated via Fourier analysis. Mean and 95% CI are computed over 100 random initializations. **(b)** Average return in DMC-Hard for 1M steps. Mean and 95% CI over 10 seeds, using SAC.

Regarding RSNorm, although its absence significantly impacts the agent's performance negatively, the overall simplicity bias score of the architecture increases rather than decreases. We speculate that this discrepancy arises because our Fourier Transform based complexity measure relies on evaluating the network's outputs over a uniformly distributed input space. RSNorm is designed to normalize activations based on the statistics of non-uniform or dynamically changing input distributions, excelling with inputs that exhibit variance and complexity not captured by a uniform distribution. Therefore, since our simplicity analysis measure is based on a uniform input distribution, it may not fully capture RSNorm's practical contribution to promoting simplicity bias.

### C.2  EXTENDED TRAINING

**Experimental Setup.** We assess SimBa's robustness by evaluating its performance on the Deep-Mind Control Suite (DMC) Hard tasks over an extended training period of 5 million timesteps. We compared three methods: SAC (Haarnoja et al., 2018), TD-MPC2 (Hansen et al., 2023), and SAC with SimBa. All models are trained under identical hyperparameters and configurations as in our main experiments to ensure a fair comparison.

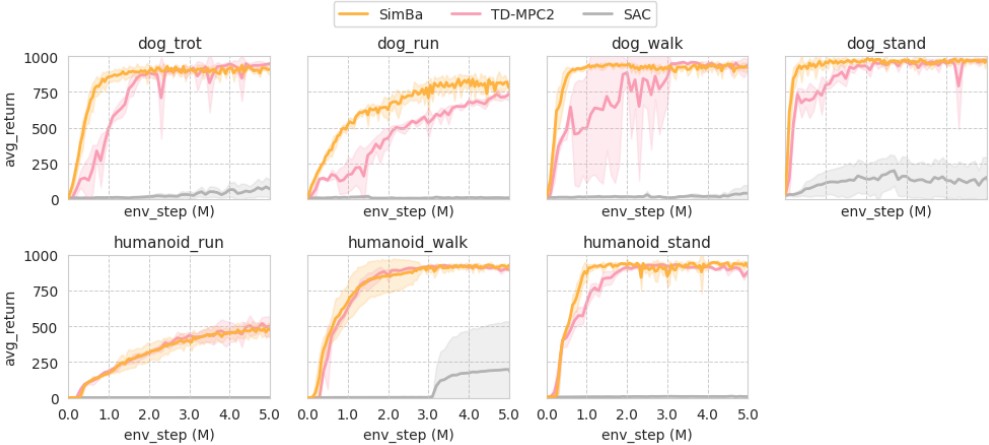

Figure 16: **Per task learning curve for DMC Hard.** Mean and 95% CI are computed over 5 seeds for SimBa, 3 seeds for TD-MPC2 and SAC.

**Results.** Figure 16 presents the learning curve of each method for 5 million timesteps. The learning curves indicate that SimBa + SAC not only learns faster but also maintains more stable and consistent performance improvements over time, demonstrating better sample efficiency and reduced variance compared to the baselines.

## D    Correlation between Simplicity Bias and Performance

To investigate the relationship between simplicity bias score and performance in deep reinforcement learning, we computed the correlation between simplicity bias scores and average returns for DMC-Hard across various architectures evaluated in our study. Our analysis includes 12 architectures: four baselines architectures-MLP, SimBa, BroNet (Nauman et al., 2024), and SpectralNet (Bjorck et al., 2021)-along with eight SimBa variants. These variants consists of four variants with component additions (Section 5.1) and four variants with component reductions (Appendix C.1), representing a gradual transition from MLP to SimBa.

As discussed in Appendix F, the current simplicity bias measure may not accurately capture the influence of RSNorm. To address this limitation, we separately evaluated the correlation for architectures incorporating RSNorm. All twelve architectures have nearly identical parameter counts, approximately 4.5 million parameters each, with a maximum variation of 1%. Performance was assessed on the DMC-Hard benchmark, trained for one million steps.

**Results.** As illustrated in Figure 17(a), architectures employing RSNorm exhibit a Pearson correlation coefficient of 0.79, indicating a strong positive correlation between simplicity bias score and average return in scaled architectures. In contrast, Figure 17(b) shows a Pearson correlation coefficient of 0.54, representing a moderate correlation across all architectures. We attribute the lower correlation in the broader set to the current limitations of the simplicity bias measure. We believe these findings support our hypothesis that higher simplicity bias scores are associated with better performance in deep reinforcement learning with scaled-up architectures.

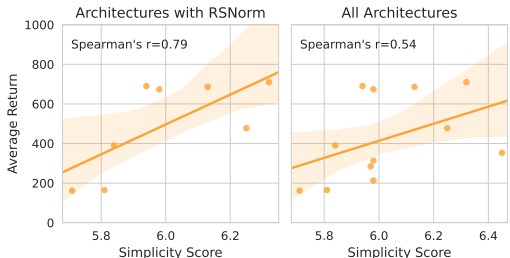

Figure 17: **Correlation Analysis.** Pearson correlation between simplicity bias score and average return on DMC-Hard. **(a)** Architectures incorporating RSNorm show a strong correlation, $\rho = 0.79$. **(b)** All evaluated architectures exhibit a moderate correlation, $\rho = 0.54$.

# E  PLASTICITY ANALYSIS

Recent studies have identified the *loss of plasticity* as a significant challenge in non-stationary training scenarios, where neural networks gradually lose their ability to adapt to new data over time (Nikishin et al., 2022; Lyle et al., 2023; Lee et al., 2024). This phenomenon can severely impact the performance and adaptability of models in dynamic learning environments such as RL. To quantify and understand this issue, several metrics have been proposed, including stable-rank (Kumar et al., 2022), dormant ratio (Sokar et al., 2023), and L2-feature norm (Kumar et al., 2022).

## E.1  METRICS

The **stable-rank** assesses the rank of the feature matrix, reflecting the diversity and richness of the representations learned by the network. This is achieved by performing an eigen decomposition on the covariance matrix of the feature matrix and counting the number of singular values $\sigma_j$ that exceed a predefined threshold $\tau$:

$$\text{s-Rank} = \sum_{j=1}^{m} \mathbb{I}(\sigma_j > \tau) \tag{16}$$

where $F \in \mathbb{R}^{d \times m}$ is the feature matrix with $d$ samples and $m$ features, $\sigma_j$ are the singular values, and $\mathbb{I}(\cdot)$ is the indicator function.

The **dormant ratio** measures the proportion of neurons with negligible activation. It is calculated as the number of neurons with activation norms below a small threshold $\epsilon$ divided by the total number of neurons $D$:

$$\text{Dormant Ratio} = \frac{|\{i \mid \|a_i\| < \epsilon\}|}{D} \tag{17}$$

where $a_i$ represents the activation of neuron $i$.

The **L2-feature norm** represents the average L2 norm of the feature vectors across all samples:

$$\text{Feature Norm} = \frac{1}{N} \sum_{i=1}^{N} \|F_i\|_2 \tag{18}$$

where $F_i$ is the feature vector for the $i$-th sample. Large feature norms can signal overactive neurons, potentially leading to numerical instability.

## E.2  RESULTS

To evaluate the impact of different architectures on network plasticity, we compared the Soft Actor-Critic (SAC) algorithm implemented with a standard Multi-Layer Perceptron (MLP) against SAC with the SimBa architecture on the DMC-Easy & Medium and DMC-Hard tasks. Additionally, we conducted an ablation study to analyze how each component of SimBa contributes to plasticity measures. Each configuration was evaluated across five random seeds.

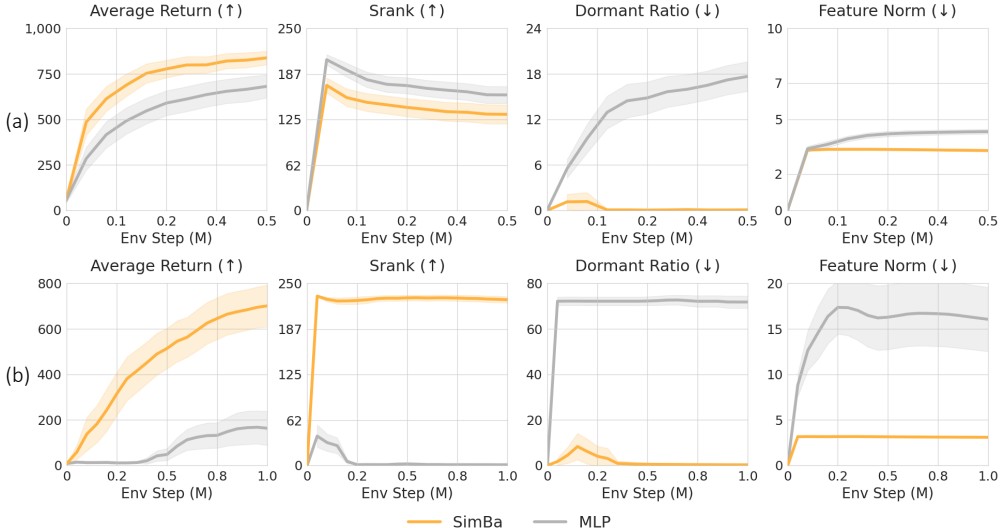

Figure 18: **Plasticity Analysis of SimBa versus MLP.** Metrics include dormant ratio, s-rank, and feature norm. Figures (a) and (b) represent DMC-Easy & Medium and DMC-Hard environments, respectively. A higher dormant ratio and feature norm, along with a lower s-rank, indicate a greater loss of plasticity.

As illustrated in Figures 18.(a) and (b), SAC with the MLP architecture exhibits high dormant ratios and large feature norms across both DMC-Easy & Medium and DMC-Hard tasks, indicating significant loss of plasticity. While MLP achieves higher s-rank values than SimBa in the DMC-Easy & Medium tasks, SimBa outperforms MLP in s-rank for the DMC-Hard tasks. Importantly, SimBa consistently maintains lower dormant ratios, balanced feature norms, and overall higher s-rank values in more complex environments.

These findings indicate that SimBa effectively preserves plasticity, avoiding the degenerative trends observed with the MLP-based SAC.

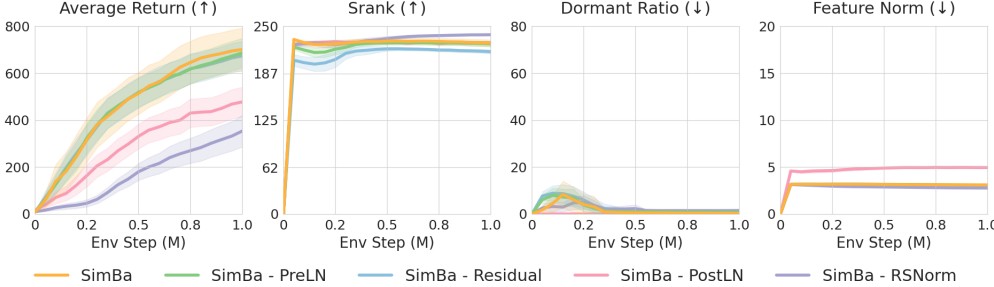

Figure 19: **Plasticity Analysis of SimBa Components.** Metrics include dormant ratio, s-rank, and feature norm. Figures (a) and (b) represent DMC-Easy & Medium and DMC-Hard environments, respectively. A higher dormant ratio and feature norm, along with a lower s-rank, indicate a greater loss of plasticity.

Furthermore, for the component ablation study (Figure 19), removing each component of SimBa individually shows diminished average returns. However, ablating each component does not significantly reduce plasticity measures, with only post-layer normalization showing an improved feature norm. This suggests that while individual components may not significantly impact plasticity measures with their sole use, their integrated use within the SimBa architecture is crucial for effectively mitigating plasticity loss.

## F    COMPARISON TO EXISTING ARCHITECTURES

Here we provide a more in-depth discussion related to the architectural difference between SimBa, BroNet, and SpectralNet. As shown in Figure 20, SimBa differs from BroNet in three key aspects: (i) the inclusion of RSNorm, (ii) the implementation of pre-layer normalization, (iii) the utilization of a linear residual pathway, (iv) and the inclusion of a post-layer normalization layer. Similarly, compared to SpectralNet, SimBa incorporates RSNorm, employs a linear residual pathway, and leverages spectral normalization differently.

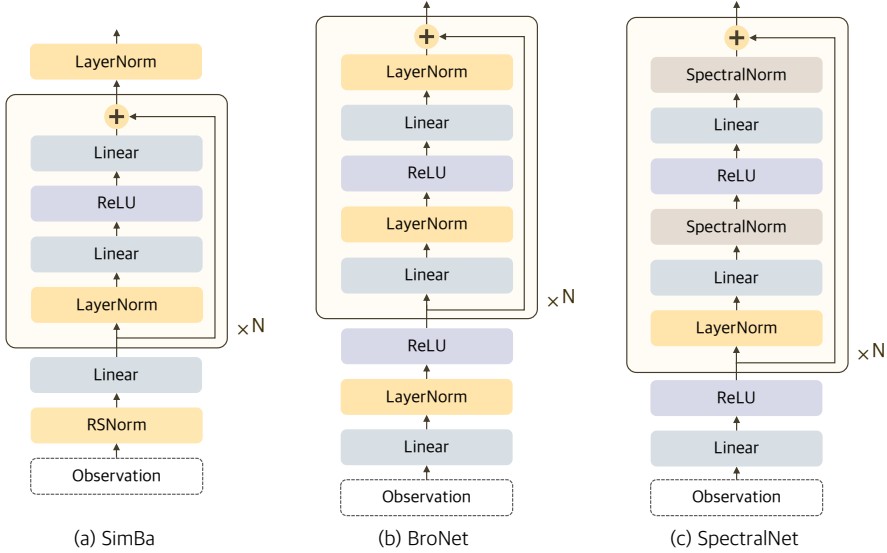

Figure 20: **Architecture Comparison.** Illustration of SimBa, BroNet, and SpectralNet.

## G    COMPUTATIONAL RESOURCES

The training was performed using NVIDIA RTX 3070, A100, or H100 GPUs for neural network computations and either a 16-core Intel i7-11800H or a 32-core AMD EPYC 7502 CPU for running simulators. Our software environment included Python 3.10, CUDA 12.2, and Jax 4.26.

When benchmarking computation time, experiments were conducted on a same hardware equipped with an NVIDIA RTX 3070 GPU and 16-core Intel i7-11800H CPU.

Table 1: **Environment details.** We list the episode length, action repeat for each domain, total environment steps, and performance metrics used for benchmarking SimBa.

|  | DMC | MyoSuite | HumanoidBench | Craftax |
|---|---|---|---|---|
| Episode length | $1,000$ | 100 | 500 - 1,000 | – |
| Action repeat | 2 | 2 | 2 | 1 |
| Effective length | 500 | 50 | 250 - 500 | – |
| Total env. steps | 500K-1M | 1M | 2M | 1B |
| Performance metric | Average Return | Average Success | Average Return | Average Score |

## H ENVIRONMENT DETAILS

This section details the benchmark environments used in our evaluation. We list all tasks from each benchmark along with their observation and action dimensions. Visualizations of each environment are provided in Figure 6, and detailed environment information is available in Table 1.

### H.1 DEEPMIND CONTROL SUITE

DeepMind Control Suite (Tassa et al., 2018, DMC) is a standard benchmark for continuous control, featuring a range of locomotion and manipulation tasks with varying complexities, from simple low-dimensional tasks ($s \in \mathbb{R}^3$) to highly complex ones ($s \in \mathbb{R}^{223}$). We evaluate 27 DMC tasks, categorized into two groups: DMC-Easy&Medium and DMC-Hard. All Humanoid and Dog tasks are classified as DMC-Hard, while the remaining tasks are grouped under DMC-Easy&Medium. The complete lists of DMC-Easy&Medium and DMC-Hard are provided in Tables 2 and 3, respectively.

For our URL experiments, we adhere to the protocol in Park et al. (2023), using an episode length of 400 steps and augmenting the agent's observations with its $x$, $y$, and $z$ coordinates. For the representation function $\phi$ (i.e., reward network) in METRA, we only used $x$, $y$, and $z$ positions as input. This approach effectively replaces the colored floors used in the pixel-based humanoid setting of METRA with state-based inputs.

### H.2 MYOSUITE

MyoSuite (Caggiano et al., 2022) simulates musculoskeletal movements with high-dimensional state and action spaces, focusing on physiologically accurate motor control. It includes benchmarks for complex object manipulation using a dexterous hand. We evaluate 10 MyoSuite tasks, categorized as "easy" when the goal is fixed and "hard" when the goal is randomized, following Hansen et al. (2023). The full list of MyoSuite tasks is presented in Table 4.

### H.3 HUMANOIDBENCH

HumanoidBench (Sferrazza et al., 2024) is a high-dimensional simulation benchmark designed to advance humanoid robotics research. It features the Unitree H1 humanoid robot with dexterous hands, performing a variety of challenging locomotion and manipulation tasks. These tasks range from basic locomotion to complex activities requiring precise manipulation. The benchmark facilitates algorithm development and testing without the need for costly hardware by providing a simulated platform. In our experiments, we focus on 14 locomotion tasks, simplifying the setup by excluding the dexterous hands. This reduces the complexity associated with high degrees of freedom and intricate dynamics. All HumanoidBench scores are normalized based on each task's target success score as provided by the authors. A complete list of tasks is available in Table 5.

### H.4 CRAFTAX

Craftax (Matthews et al., 2024) is an open-ended RL environment that combines elements from Crafter (Hafner, 2022) and NetHack (Küttler et al., 2020). It presents a challenging scenario requiring the sequential completion of numerous tasks. A key feature of Craftax is its support for vectorized parallel environments and a full end-to-end GPU learning pipeline, enabling a large number

of environment steps at low computational cost. The original baseline performance in Craftax is reported as a percentage of the maximum score (226). In our experiments, we report agents' raw average scores instead.

Table 2: **DMC Easy & Medium.** We consider a total of 20 continuous control tasks for the DMC Easy & Medium benchmark. We list all considered tasks below and baseline performance for each task is reported at 500K environment steps.

| Task | Observation dim | Action dim |
|------|-----------------|------------|
| Acrobot Swingup | 6 | 1 |
| Cartpole Balance | 5 | 1 |
| Cartpole Balance Sparse | 5 | 1 |
| Cartpole Swingup | 5 | 1 |
| Cartpole Swingup Sparse | 5 | 1 |
| Cheetah Run | 17 | 6 |
| Finger Spin | 9 | 2 |
| Finger Turn Easy | 12 | 2 |
| Finger Turn Hard | 12 | 2 |
| Fish Swim | 24 | 5 |
| Hopper Hop | 15 | 4 |
| Hopper Stand | 15 | 4 |
| Pendulum Swingup | 3 | 1 |
| Quadruped Run | 78 | 12 |
| Quadruped Walk | 78 | 12 |
| Reacher Easy | 6 | 2 |
| Reacher Hard | 6 | 2 |
| Walker Run | 24 | 6 |
| Walker Stand | 24 | 6 |
| Walker Walk | 24 | 6 |

Table 3: **DMC Hard.** We consider a total of 7 continuous control tasks for the DMC Hard benchmark. We list all considered tasks below and baseline performance for each task is reported at 1M environment steps.

| Task | Observation dim | Action dim |
|------|-----------------|------------|
| Dog Run | 223 | 38 |
| Dog Trot | 223 | 38 |
| Dog Stand | 223 | 38 |
| Dog Walk | 223 | 38 |
| Humanoid Run | 67 | 24 |
| Humanoid Stand | 67 | 24 |
| Humanoid Walk | 67 | 24 |

Table 4: **MyoSuite.** We consider a total of 10 continuous control tasks for the MyoSuite benchmark, which encompasses both fixed-goal ('Easy') and randomized-goal ('Hard') settings. We list all considered tasks below and baseline performance for each task is reported at 1M environment steps.

| Task | Observation dim | Action dim |
|------|-----------------|------------|
| Key Turn Easy | 93 | 39 |
| Key Turn Hard | 93 | 39 |
| Object Hold Easy | 91 | 39 |
| Object Hold Hard | 91 | 39 |
| Pen Twirl Easy | 83 | 39 |
| Pen Twirl Hard | 83 | 39 |
| Pose Easy | 108 | 39 |
| Pose Hard | 108 | 39 |
| Reach Easy | 115 | 39 |
| Reach Hard | 115 | 39 |

Table 5: **HumanoidBench.** We consider a total of 14 continuous control locomotion tasks for the UniTree H1 humanoid robot from the HumanoidBench domain. We list all considered tasks below and baseline performance for each task is reported at 2M environment steps.

| Task | Observation dim | Action dim |
|------|-----------------|------------|
| Balance Hard | 77 | 19 |
| Balance Simple | 64 | 19 |
| Crawl | 51 | 19 |
| Hurdle | 51 | 19 |
| Maze | 51 | 19 |
| Pole | 51 | 19 |
| Reach | 57 | 19 |
| Run | 51 | 19 |
| Sit Simple | 51 | 19 |
| Sit Hard | 64 | 19 |
| Slide | 51 | 19 |
| Stair | 51 | 19 |
| Stand | 51 | 19 |
| Walk | 51 | 19 |

Table 6: **Craftax.** We consider the symbolic version of Craftax. Baseline performance is reported at 1B environment steps.

| Task | Observation dim | Action dim |
|------|-----------------|------------|
| Craftax-Symbolic-v1 | 8268 | 43 |

# I  HYPERPARAMETERS

## I.1  OFF-POLICY

Table 7: **SAC+SimBa hyperparameters.** The hyperparameters listed below are used consistently across all tasks when integrating SimBa with SAC. For the discount factor, we set it automatically using heuristics used by TD-MPC2 (Hansen et al., 2023).

| Hyperparameter | Value |
|---|---|
| Critic block type | SimBa Residual |
| Critic num blocks | 2 |
| Critic hidden dim | 512 |
| Critic learning rate | 1e-4 |
| Target critic momentum ($\tau$) | 5e-3 |
| Actor block type | SimBa Residual |
| Actor num blocks | 1 |
| Actor hidden dim | 128 |
| Actor learning rate | 1e-4 |
| Initial temperature ($\alpha_0$) | 1e-2 |
| Temperature learning rate | 1e-4 |
| Target entropy ($\mathcal{H}^*$) | $|\mathcal{A}|/2$ |
| Batch size | 256 |
| Optimizer | AdamW |
| Optimizer momentum ($\beta_1, \beta_2$) | (0.9, 0.999) |
| Weight decay ($\lambda$) | 1e-2 |
| Discount ($\gamma$) | Heuristic |
| Replay ratio | 2 |
| Clipped Double Q | HumanoidBench: True |
| | Other Envs: False |

Table 8: **DDPG+SimBa hyperparameters.** The hyperparameters listed below are used consistently across all tasks when integrating SimBa with DDPG. For the discount factor, we set it automatically using heuristics used by TD-MPC2 (Hansen et al., 2023).

| Hyperparameter | Value |
|---|---|
| Critic block type | SimBa Residual |
| Critic num blocks | 2 |
| Critic hidden dim | 512 |
| Critic learning rate | 1e-4 |
| Target critic momentum ($\tau$) | 5e-3 |
| Actor block type | SimBa Residual |
| Actor num blocks | 1 |
| Actor hidden dim | 128 |
| Actor learning rate | 1e-4 |
| Exploration noise | $\mathcal{N}(0, 0.1^2)$ |
| Batch size | 256 |
| Optimizer | AdamW |
| Optimizer momentum ($\beta_1, \beta_2$) | (0.9, 0.999) |
| Weight decay ($\lambda$) | 1e-2 |
| Discount ($\gamma$) | Heuristic |
| Replay ratio | 2 |

Table 9: **TDMPC2+SimBa hyperparameters.** We provide a detailed list of the hyperparameters used for the shared encoder module in TD-MPC2 (Hansen et al., 2023). Aside from these, we follow the hyperparameters specified in the original TD-MPC2 paper. The listed hyperparameters are applied uniformly across all tasks when integrating SimBa with TD-MPC2.

| Hyperparameter | Value |
|---|---|
| Encoder block type | SimBa Residual |
| Encoder num blocks | 2 |
| Encoder hidden dim | 256 |
| Encoder learning rate | 1e-4 |
| Encoder weight decay | 1e-1 |

## I.2 ON-POLICY

Table 10: **PPO+SimBa hyperparameters.** We provide a detailed list of the hyperparameters when integrating SimBa with PPO (Schulman et al., 2017). Aside from these, we follow the hyperparameters specified in the Craftax paper (Matthews et al., 2024).

| Hyperparameter | Value |
|---|---|
| Critic block type | SimBa Residual |
| Critic num blocks | 2 |
| Critic hidden dim | 512 |
| Critic learning rate | 1e-4 |
| Actor block type | SimBa Residual |
| Actor num blocks | 1 |
| Actor hidden dim | 256 |
| Actor learning rate | 1e-4 |
| Optimizer | AdamW |
| Optimizer momentum $(\beta_1, \beta_2)$ | (0.9, 0.999) |
| Optimizer eps | 1e-8 |
| Weight decay $(\lambda)$ | 1e-2 |

## I.3 UNSUPERVISED RL

Table 11: **METRA+SimBa hyperparameters.** We provide a detailed list of the hyperparameters when integrating SimBa with METRA (Park et al., 2023). Aside from these, we follow the hyperparameters specified in the original METRA paper.

| Hyperparameter | Value |
|---|---|
| Critic block type | SimBa Residual |
| Critic num blocks | 2 |
| Critic hidden dim | 512 |
| Actor block type | SimBa Residual |
| Actor num blocks | 1 |
| Actor hidden dim | 128 |

## J  LEARNING CURVE

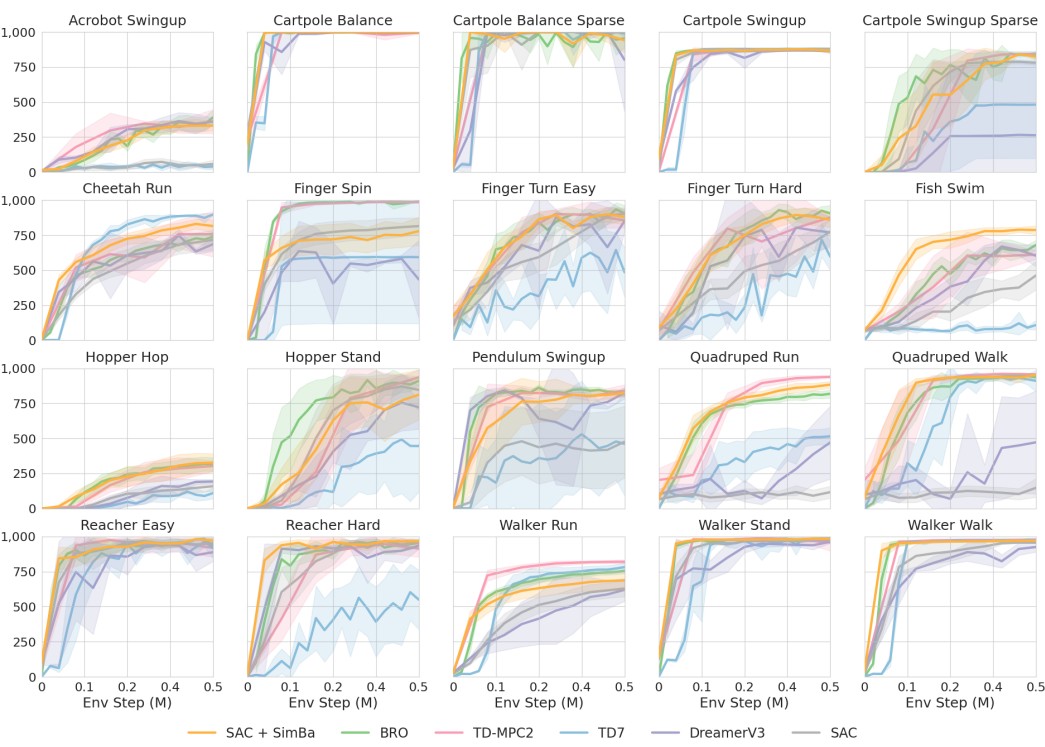

Figure 21: **Per task learning curve for DMC Easy&Medium.** Mean and 95% CI over 10 seeds for SimBa and BRO, 5 seeds for TD7 and SAC, 3 seeds for TD-MPC2 and DreamerV3.

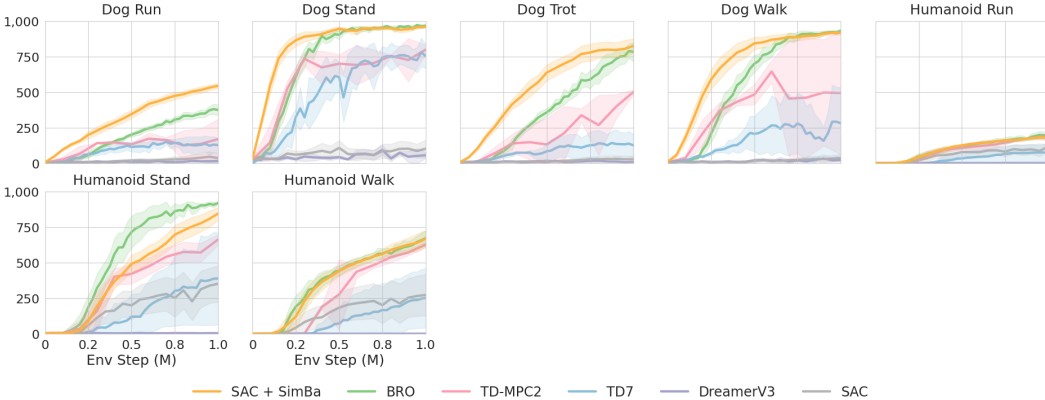

Figure 22: **Per task learning curve for DMC Hard.** Mean and 95% CI over 10 seeds for SimBa and BRO, 5 seeds for TD7 and SAC, 3 seeds for TD-MPC2 and DreamerV3.

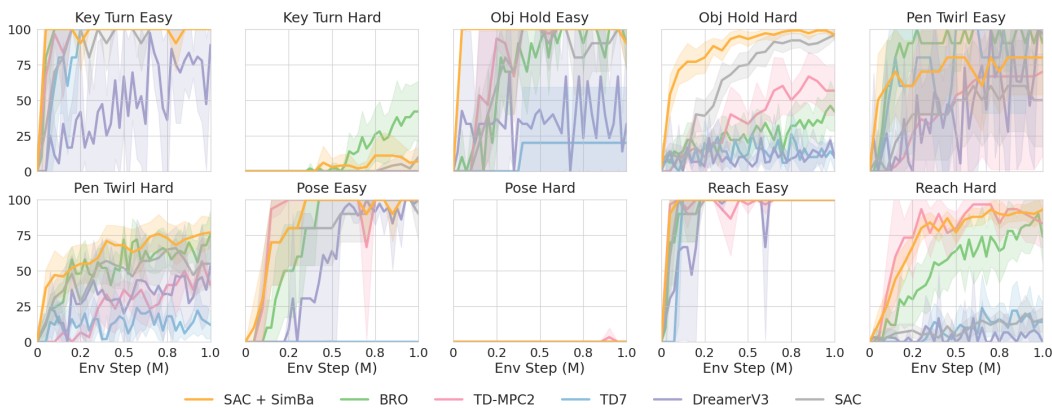

Figure 23: **Per task learning curve for MyoSuite.** Mean and 95% CI over 10 seeds for SimBa and BRO, 5 seeds for TD7 and SAC, 3 seeds for TD-MPC2 and DreamerV3.

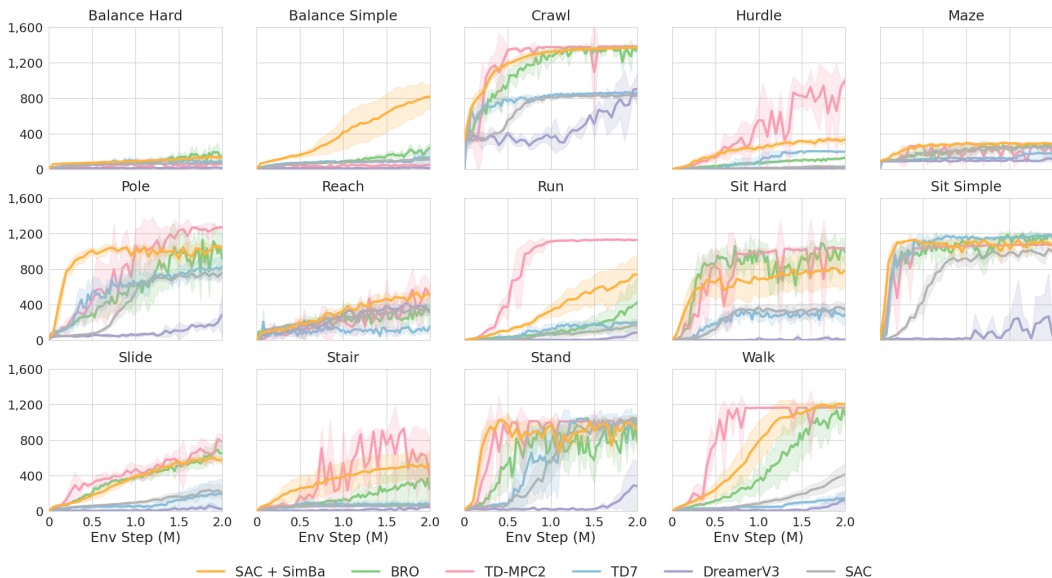

Figure 24: **Per task learning curve for HumanoidBench.** Mean and 95% CI over 10 seeds for SimBa, 5 seeds for BRO, TD7, and SAC, 3 seeds for TD-MPC2 and DreamerV3. We no

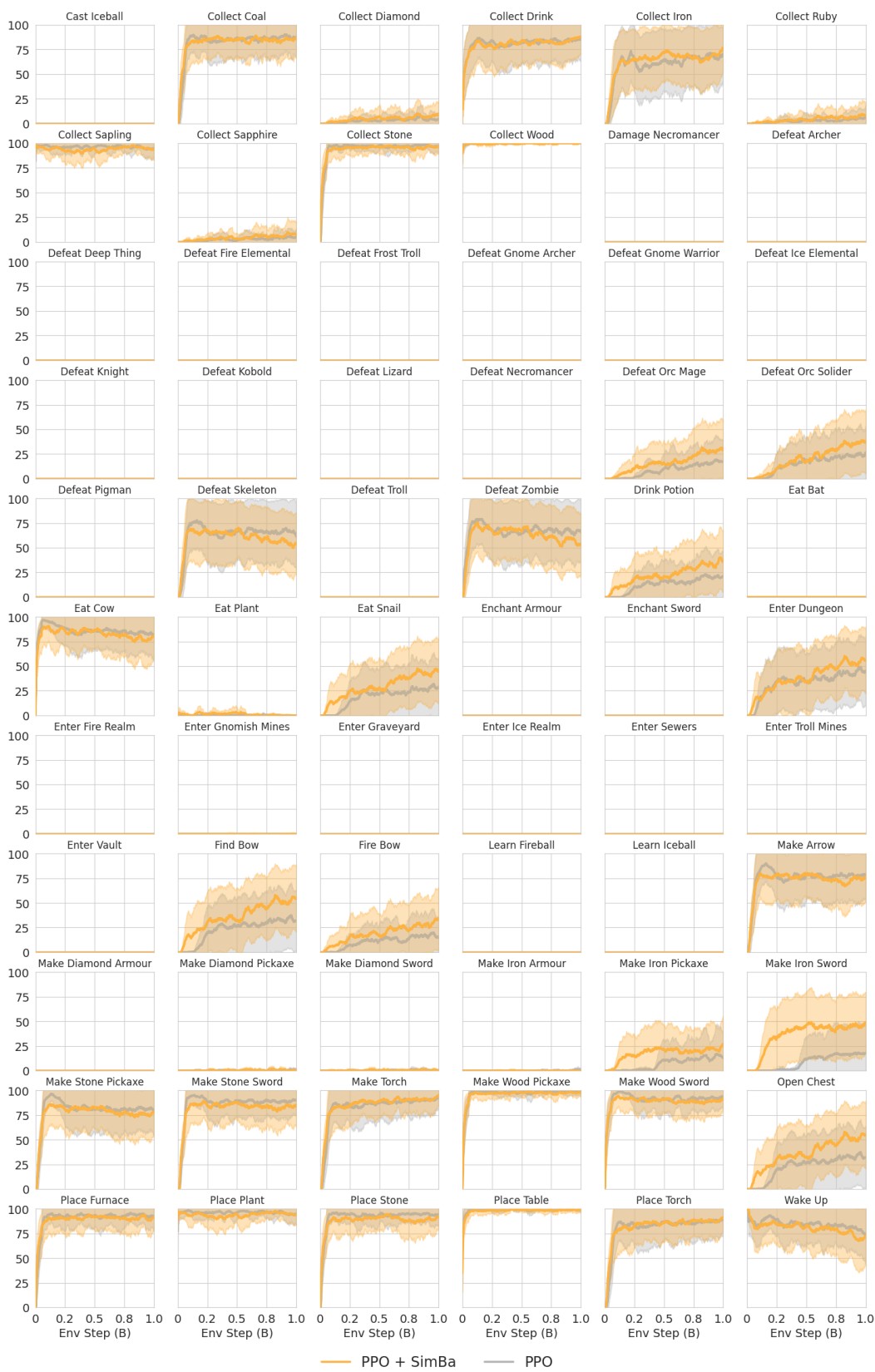

Figure 25: **Per task learning curve for Craftax.** We visualize the success rate learning curve for 66 tasks in Craftax. Mean and 95% CI over 5 seeds for PPO + SimBa and PPO.

# K   FULL RESULT

Table 12: **Per task results for DMC Easy&Medium.** Results for SimBa and BRO are averaged over 10 seeds, for TD7 and SAC over 5 seeds, and for TD-MPC2 and DreamerV3 over 3 seeds.

| Method | SimBa | BRO | TD7 | SAC | TD-MPC2 | DreamerV3 |
|---|---|---|---|---|---|---|
| Acrobot Swingup | 331.57 | 390.78 | 39.83 | 57.56 | 361.07 | 360.46 |
| Cartpole Balance | 999.05 | 998.66 | 998.79 | 998.79 | 993.93 | 994.95 |
| Cartpole Balance Sparse | 940.53 | 954.21 | 988.58 | 1000.00 | 1000.00 | 800.25 |
| Cartpole Swingup | 866.53 | 878.69 | 878.09 | 863.24 | 876.07 | 863.90 |
| Cartpole Swingup Sparse | 823.97 | 833.18 | 480.74 | 779.99 | 844.77 | 262.69 |
| Cheetah Run | 814.97 | 739.67 | 897.76 | 716.43 | 757.60 | 686.25 |
| Finger Spin | 778.83 | 987.45 | 592.74 | 814.69 | 984.63 | 434.06 |
| Finger Turn Easy | 881.33 | 905.85 | 485.66 | 903.07 | 854.67 | 851.11 |
| Finger Turn Hard | 860.22 | 905.30 | 596.32 | 775.21 | 876.27 | 769.86 |
| Fish Swim | 786.73 | 680.34 | 108.84 | 462.67 | 610.23 | 603.34 |
| Hopper Hop | 326.69 | 315.04 | 110.27 | 159.43 | 303.27 | 192.70 |
| Hopper Stand | 811.75 | 910.88 | 445.50 | 845.89 | 936.47 | 722.42 |
| Pendulum Swingup | 824.53 | 816.20 | 461.40 | 476.58 | 841.70 | 825.17 |
| Quadruped Run | 883.68 | 818.62 | 515.13 | 116.91 | 939.63 | 471.25 |
| Quadruped Walk | 952.96 | 936.06 | 910.55 | 147.83 | 957.17 | 472.31 |
| Reacher Easy | 972.23 | 933.77 | 920.38 | 951.80 | 919.43 | 888.36 |
| Reacher Hard | 965.96 | 956.52 | 549.50 | 959.59 | 913.73 | 935.25 |
| Walker Run | 687.16 | 754.43 | 782.32 | 629.44 | 820.40 | 620.13 |
| Walker Stand | 983.02 | 986.58 | 984.63 | 972.59 | 957.17 | 963.28 |
| Walker Walk | 970.73 | 973.41 | 976.58 | 956.67 | 978.70 | 925.46 |
| IQM | 885.70 | 888.02 | 740.19 | 799.75 | 895.47 | 748.58 |
| Median | 816.78 | 836.62 | 623.18 | 676.59 | 836.93 | 684.18 |
| Mean | 823.12 | 833.78 | 636.18 | 679.42 | 836.34 | 682.16 |
| OG | 0.1769 | 0.1662 | 0.3638 | 0.3206 | 0.1637 | 0.3178 |

Table 13: **Per task results for DMC Hard.** Results for SimBa and BRO are averaged over 10 seeds, for TD7 and SAC over 5 seeds, and for TD-MPC2 and DreamerV3 over 3 seeds.

| Method | SimBa | BRO | TD7 | SAC | TD-MPC2 | DreamerV3 |
|---|---|---|---|---|---|---|
| Dog Run | 544.86 | 374.63 | 127.48 | 36.86 | 169.87 | 15.72 |
| Dog Stand | 960.38 | 966.97 | 753.23 | 102.04 | 798.93 | 55.87 |
| Dog Trot | 824.69 | 783.12 | 126.00 | 29.36 | 500.03 | 10.19 |
| Dog Walk | 916.80 | 931.46 | 280.87 | 38.14 | 493.93 | 23.36 |
| Humanoid Run | 181.57 | 204.96 | 79.32 | 116.97 | 184.57 | 0.91 |
| Humanoid Stand | 846.11 | 920.11 | 389.80 | 352.72 | 663.73 | 5.12 |
| Humanoid Walk | 668.48 | 672.55 | 252.72 | 273.67 | 628.23 | 1.33 |
| IQM | 773.28 | 771.50 | 216.04 | 69.03 | 527.11 | 9.63 |
| Median | 706.39 | 694.20 | 272.62 | 159.36 | 528.26 | 17.13 |
| Mean | 706.13 | 693.40 | 287.06 | 135.68 | 491.33 | 16.07 |
| OG | 0.2939 | 0.3066 | 0.7129 | 0.8643 | 0.5087 | 0.9839 |

Table 14: **Per task results for MyoSuite.** Results for SimBa and BRO are averaged over 10 seeds, for TD7 and SAC over 5 seeds, and for TD-MPC2 and DreamerV3 over 3 seeds.

| Method | SimBa | BRO | TD7 | SAC | TD-MPC2 | DreamerV3 |
|---|---|---|---|---|---|---|
| Key Turn Easy | 100.00 | 100.00 | 100.00 | 100.00 | 100.00 | 88.89 |
| Key Turn Hard | 7.00 | 42.00 | 0.00 | 10.00 | 0.00 | 0.00 |
| Object Hold Easy | 90.00 | 90.00 | 20.00 | 90.00 | 100.00 | 33.33 |
| Object Hold Hard | 96.00 | 42.00 | 10.00 | 96.00 | 56.67 | 9.44 |
| Pen Twirl Easy | 80.00 | 90.00 | 100.00 | 50.00 | 70.00 | 96.67 |
| Pen Twirl Hard | 77.00 | 76.00 | 12.00 | 55.00 | 40.00 | 53.33 |
| Pose Easy | 100.00 | 100.00 | 0.00 | 90.00 | 100.00 | 100.00 |
| Pose Hard | 0.00 | 0.00 | 0.00 | 0.00 | 0.00 | 0.00 |
| Reach Easy | 100.00 | 100.00 | 100.00 | 100.00 | 100.00 | 100.00 |
| Reach Hard | 93.00 | 74.00 | 14.00 | 16.00 | 83.33 | 0.00 |
| IQM | 95.20 | 87.60 | 22.31 | 71.40 | 77.50 | 46.56 |
| Median | 77.00 | 72.00 | 34.00 | 62.50 | 65.00 | 47.00 |
| Mean | 74.30 | 71.40 | 35.60 | 60.70 | 65.00 | 48.17 |
| OG | 99.93 | 99.93 | 99.96 | 99.94 | 99.94 | 99.95 |

Table 15: **Per task results for HumanoidBench.** Results for SimBa are averaged over 10 seeds, for BRO, TD7 and SAC over 5 seeds, and for TD-MPC2 and DreamerV3 over 3 seeds.

| Method | SimBa | BRO | TD7 | SAC | TD-MPC2 | DreamerV3 |
|---|---|---|---|---|---|---|
| Balance Hard | 137.20 | 145.95 | 79.90 | 69.02 | 64.56 | 16.07 |
| Balance Simple | 816.38 | 246.57 | 132.82 | 113.38 | 50.69 | 14.09 |
| Crawl | 1370.51 | 1373.83 | 868.63 | 830.56 | 1384.40 | 906.66 |
| Hurdle | 340.60 | 128.60 | 200.28 | 31.89 | 1000.12 | 18.78 |
| Maze | 283.58 | 259.03 | 179.23 | 254.38 | 198.65 | 114.90 |
| Pole | 1036.70 | 915.89 | 830.72 | 760.78 | 1269.57 | 289.18 |
| Reach | 523.10 | 317.99 | 159.37 | 347.92 | 505.61 | 341.62 |
| Run | 741.16 | 429.74 | 196.85 | 168.25 | 1130.94 | 85.93 |
| Sit Hard | 783.95 | 989.07 | 293.96 | 345.65 | 1027.47 | 9.95 |
| Sit Simple | 1059.73 | 1151.84 | 1183.26 | 994.58 | 1074.96 | 40.53 |
| Slide | 577.87 | 653.17 | 197.07 | 208.46 | 780.82 | 24.43 |
| Stair | 527.49 | 249.86 | 77.19 | 65.53 | 398.21 | 49.04 |
| Stand | 906.94 | 780.52 | 1005.54 | 1029.78 | 1020.59 | 280.99 |
| Walk | 1202.91 | 1080.63 | 143.86 | 412.21 | 1165.42 | 125.59 |
| IQM | 747.43 | 558.03 | 256.77 | 311.46 | 885.67 | 72.30 |
| Median | 733.09 | 632.93 | 385.41 | 399.93 | 796.18 | 160.49 |
| Mean | 736.30 | 623.05 | 396.33 | 402.31 | 787.64 | 165.55 |
| OG | 0.3197 | 0.4345 | 0.6196 | 0.6016 | 0.2904 | 0.8352 |

