# OpenReview forum: "SimBa: Simplicity Bias for Scaling Up Parameters in Deep Reinforcement Learning"
_ICLR.cc/2025/Conference — ICLR 2025 Spotlight_

### Official Review · Reviewer_8rZo · 2024-10-23

**Soundness:** 3
**Presentation:** 3
**Contribution:** 4
**Rating:** 8
**Confidence:** 5

**Summary:**

This paper builds upon the insights of the paper by [Teney et al, 2024], which shows that certain architectural implementations provide a simplicity bias, which could be one of the more causal explanations to their practical usefulness.

In order to transfer this notion of simplicity bias to RL, the authors run tests with SAC on proprietary state-based RL, ranging from easy (Cartpole etc.) to hard (Humanoid, Dog, etc.) environments.

To maximize the simplicity bias of vanilla SAC, the authors augment the original architecture with RSnorm, Residual Feedforward blocks and Layer Normalization. Additionally, the authors scale up the architecture.

The results are promising, showing that keeping a simplicity bias in the network allows for scaling up the network, elevating vanilla SAC to the level where it can perform very well on DMC-Hard tasks, without using algorithmic advancements.

Although results are only done on state-based architectures, the harder environments like Humanoid & Dog still allow for sufficient difficulty in the evaluation. Additionally, pixel-based environments have already shown correlation between simplicity bias in the CNN and performance (Residual connections in CNN’s -> Impala architecture), although to the best of my knowledge, the reason being a simplicity bias was never concluded by these researchers.

**Strengths:**

- Easy to implement, which is promising for future work.

- Good comparison with a SOTA algorithm TD-MPC2, and even an ablation using SimBa on TD-MPC2.

- A strong step forward in the domain of non-algorithmic improvements in RL.

**Weaknesses:**

- The plasticity analysis in Appendix C. could be confusing to the average reader. It shows a comparison to a basically ‘collapsed’ baseline MLP, as the effective rank is negligible. This might give a reader the impression that these are the representative network properties of training with a vanilla MLP, and paints a too strong picture of SimBa’s added network properties. I believe it would therefore be much more informative to the reader to add an additional 2 rows of figures to Fig. 15:
 		- Row 1: Plasticity analysis on DMC - medium
                - Row 2: Plasticity analysis with all the ablations of Fig. 4, to show the unique effects of every module.

- As important as it is to see that SimBa is computationally cheap, it would be also interesting to add some figures showing longer training (e.g. 5 million timesteps) on DMC hard and comparing to TD-MPC2 and the baseline SAC.

- I believe a mention of the improvements that the Impala network gave to RL should be in the paper, and maybe a short sentence about the correlation (Residuals connections -> Simplicity Bias -> better Convergence).

**Questions:**

See Weaknesses section.

---

> ### Author Response · Authors · 2024-11-20
>
> Dear reviewer 8rZo,
>
> Thank you for your constructive feedback. To address your concerns, we have provided a detailed response to each comment. In addition, we have provided a common response to address key feedback from all reviewers. Please let us know if you have any further comments or feedback.
>
>
> > **Question 4.1**
> "The plasticity analysis in Appendix C. could be confusing to the average reader. It shows a comparison to a basically ‘collapsed’ baseline MLP, as the effective rank is negligible. I believe it would therefore be much more informative to the reader to add an additional 2 rows of figures to Fig. 15:
> > - Row 1: Plasticity analysis on DMC - medium
> > - Row 2: Plasticity analysis with all the ablations of Fig. 4, to show the unique effects of every module."
>
>
> We appreciate your suggestion to enhance the plasticity analysis for clarity. In response, we have conducted the following additional experiments:
>
> - Added a plasticity analysis on the DMC-Medium environments.
> - Included a plasticity analysis with all ablations from Fig. 4 to demonstrate the unique effects of each module.
>
> These additions have been incorporated into **Fig. 17 and 18 in Appendix C**.
>
> For the analysis on the DMC-Easy\&Medium environment, SAC with the MLP architecture still exhibit high dormant ratios and large feature norms across both DMC-Easy \& Medium indicating greater loss of plasticity.
>
> For the component ablation study, integrating each component of SimBa individually shows improved average returns. However, ablating each component does not lead to significant reductions in plasticity measures, with only Post Layer Normalization showing an improved feature norm. This suggests that while individual components may not significantly impact plasticity measures on their own, their integration within the SimBa architecture is crucial for effectively mitigating plasticity loss.
>
>
>
> > **Question 4.2**
> As important as it is to see that SimBa is computationally cheap, it would be also interesting to add some figures showing longer training (e.g. 5 million timesteps) on DMC hard and comparing to TD-MPC2 and the baseline SAC.
>
> Thank you for highlighting the importance of evaluating SimBa over extended training periods. We have expanded our experiments to include training for 5 million timesteps on DMC-Hard environments. The results are presented in Appendix C under the additional ablations section.
>
> | Training Steps | SAC | TD-MPC2 | SimBa + SAC |
> |----------------|-----|---------|-------------|
> | 1M             | 22  | 491     | 710         |
> | 5M             | 68  | 839     | 843         |
>
>
> The learning curves indicate that SimBa + SAC not only learns faster but also maintains more stable and consistent performance improvements over time, demonstrating better sample efficiency and reduced variance compared to the baselines.
>
> Thanks to the reviewer's feedback, our new experiment confirm SimBa’s robustness and effectiveness in long-term training scenarios.
>
>
> > **Question 4.3**
> I believe a mention of the improvements that the Impala network gave to RL should be in the paper, and maybe a short sentence about the correlation (Residuals connections -> Simplicity Bias -> better Convergence).
>
> We agree with your suggestion and have updated the Related Work section on page 4 to include a citation of the Impala network.

---

> > ### Comment · Reviewer_8rZo · 2024-11-22
> > **Reply to Author response**
> >
> > Thanks for the changes. I believe that, although the added plasticity analysis for DMC Easy and Medium does not have the same results as for DMC hard, it does make the paper more realistic and informative to the reader. I have no further questions. However, I personally think the scores on the extended training are also quite good, and might even be suitable for the main text. But I will leave this decision to you.
> >
> > I will keep my score of 8.

---

> > > ### Author Response · Authors · 2024-11-23
> > >
> > > Thank you so much your feedback and positive supports! We will consider adding the extended training experiment to the main text in the later version.

---

### Official Review · Reviewer_2Hpz · 2024-10-29

**Soundness:** 3
**Presentation:** 4
**Contribution:** 3
**Rating:** 8
**Confidence:** 4

**Summary:**

The authors propose the SimBa architecture, which, by inducing simplicity bias, allows scaling the size of neural networks in deep reinforcement learning problems. The proposed architecture is a neural network with residual connections, with post-layer normalization and observation normalization at the beginning. The authors claim that SimBa enables networks to scale up without sacrificing generalizability, avoiding the overfitting issues often associated with high-parameter models in RL.

**Strengths:**

1. Clear and Well-Motivated Objective: The paper identifies simplicity bias as an underexplored factor in RL network scaling.
2. Reproducibility Efforts: A public codebase and descriptions of evaluation setups are provided.
3. Broad experimental setup.

**Weaknesses:**

1. Statistical significance is not apparent from the presented results. The standard error overlaps in some plots, e.g., Figures 1b, BRO, and SimBa on Figure 5b, and Figure 8 has no rages, but I guess it would be more challenging to add them since the analysis is in two dimensions, and also from Figure 14. The tables in the Appendix do not have standard deviations. Moreover, authors could use some t-test or Mann–Whitney U test to demonstrate statistical significance.
2. It would be valuable to see reversed to Fig. 4 ablation, i.e., simplicity and Return when you turn off every single component. Clearly, all components analyzed in section 5.1 have some synergic effect -- they do not help much alone. However, it is not apparent how all of them are important in this synergy, i.e., from section 7.1, we know that RSNorm is very important -- If I understand correctly, SimBa without observation normalization works very poorly. But how will the lack of RSNorm affect the simplicity of the architecture? -- I would suggest visualizing this similarly to Figure 4, but on the Y-axis, it would be, e.g., SimBa - RSNorm or SimBa - Residual.

**Questions:**

1. Out of curiosity, SimBa is usually the most time-efficient (except SAC), but on HumanoidBench, it is slightly worse than TD7 and BRO. Do you know why?
2. The simplicity bias hypothesis looks pretty general; therefore, how will this architecture work on image-based problems?
3.  How do you expect SimBa to be applied in the discrete control tasks? Will it be enough to beat BBF on Atari as BRO on continuous control? -- Of course, new experiments on Atari would be too much for the rebuttal period, but maybe the authors have some thoughts about this.
4. How do your results about simplicity bias measured through Fourier features relate to this paper [1]? As I understand, authors of [1] claim that neural networks are naturally biased to learn low frequencies faster, and it is hard for them to learn high-frequency signals, which is present in reinforcement learning. SimBa promotes architectures that are biased towards low frequencies. Could you share your perspective on how these findings interact or complement each other?

[1] Yang, G., Ajay, A., & Agrawal, P. (2022). Overcoming the spectral bias of neural value approximation. arXiv preprint arXiv:2206.04672.

---

> ### Author Response · Authors · 2024-11-20
>
> Dear reviewer 2Hpz,
>
> Thank you for your insightful and constructive comment. Your thorough review of each section of the paper has significantly improved its overall quality. In addition, we have provided a common response to address key feedback from all reviewers.  Below, we have provided responses to each of your questions. Please let us know if you have any other concerns.
>
>
> > **Question 3.1**
> Statistical significance is not apparent from the presented results. The standard error overlaps in some plots, e.g., Figures 1b, BRO, and SimBa on Figure 5b, and Figure 8 has no rages, but I guess it would be more challenging to add them since the analysis is in two dimensions, and also from Figure 14. The tables in the Appendix do not have standard deviations. Moreover, authors could use some t-test or Mann–Whitney U test to demonstrate statistical significance.
>
> We appreciate the reviewer’s feedback regarding statistical significance and its importance in validating our results. To address this, we performed a Mann–Whitney U Test to assess the significance of the observed differences between SimBa and BRO across all environments. The results are summarized in the table below:
>
> | Environment            | avg. simba | avg. BRO | Mann-Whitney U Test (p-value)|
> |------------------------|------------|----------|------------------------------|
> | DMC Easy & Medium      | 823.12     | 833.78   | 0.089                        |
> | DMC Hard               | 705.33     | 693.40   | 0.571                        |
> | Myosuite               | 0.743      | 0.714    | 0.037                        |
> | HumanoidBench          | 732.52     | 623.05   | 0.008                        |
>
> From the results, the performance differences between SimBa and BRO are statistically significant for Myosuite and HumanoidBench ($p$ < 0.05), highlighting SimBa's effectiveness compared to BRO in these environments. In contrast, the differences in the DMC Easy & Medium and DMC Hard environments are not statistically significant, as indicated by the higher p-values.
>
> However, we would like to highlight that SimBa's strengths are not limited to achieving high performance for diverse tasks. Its design emphasizes computational efficiency and simplicity, setting it apart from BRO. Unlike BRO, which incorporates RL-specific techniques such as periodic resets, quantile regression losses, and complex explortaion techniques, SimBa achieves competitive results solely through architectural enhancements. This approach avoids any modification to the learning objective or loss function of the underlying RL algorithm, ensuring that SimBa remains both straightforward to implement and broadly applicable across diverse scenarios.
>
>
>
> > **Question 3.2**
> It would be valuable to see reversed to Fig. 4 ablation, i.e., simplicity and Return when you turn off every single component. I would suggest visualizing this similarly to Figure 4.
>
>
> We appreciate your observation regarding the importance of providing experimental results for the removal of each component in SimBa. To address this, we conducted additional ablation studies and included the following analyses:
>
> - Evaluated the impact of removing individual components from SimBa on mean performance for DMC Hard.
> - Assessed the simplicity measure of the architecture after the removal of each individual component from SimBa.
>
> We have included these results in our revised version of the manuscript in **Appendix C.1** and are illustrated in **Figure 15**.
>
> As shown in Figure 15, the removal of any single component results in a degradation of overall performance, underscoring the necessity of each component for SimBa’s effectiveness. Additionally, we observe that, except for RSNorm, the removal of residual connections, pre-layer normalizations, or post-layer normalizations leads to a reduction in the simplicity score. This further emphasizes the critical role of each individual component in SimBa’s design.
>
> Regarding RSNorm, although its removal significantly impacts the agent's performance, the overall simplicity score of the architecture increases rather than decrease. We believe this discrepancy arises because our Fourier Transform-based complexity measure heavily relies on evaluating the network's outputs over a uniformly distributed input space. RSNorm is specifically designed to normalize activations based on the statistics of non-uniform or dynamically changing input distributions, excelling with inputs that exhibit variance and complexity not captured by a uniform distribution. Therefore, using a uniform input distribution in our simplicity analysis does not fully showcase RSNorm's advantages or accurately reflect its impact on the simplicity score.

---

> > ### Author Response · Authors · 2024-11-20
> >
> > > **Question 3.3**
> > Out of curiosity, SimBa is usually the most time-efficient (except SAC), but on HumanoidBench, it is slightly worse than TD7 and BRO. Do you know why?
> >
> > Thank you for highlighting this observation. As detailed in Appendix H (page 26), we applied CDQ to SimBa on HumanoidBench, which increased the computational cost. Although our testing was not exhaustive, we observed the following:
> >
> > - Non-Episodic Tasks: CDQ had no significant impact on performance.
> > - Episodic Tasks: CDQ improved SimBa's performance.
> >
> > We hypothesize that in episodic tasks, accurately assigning or estimating values is crucial to prevent catastrophic failures and terminations. Preliminary experiments on Mujoco environments with terminations support this hypothesis, demonstrating that CDQ enhances performance in such settings. We believe these results open up an exciting direction for future research.
> >
> >
> >
> > > **Question 3.4**
> > The simplicity bias hypothesis looks pretty general; therefore, how will this architecture work on image-based problems?
> >
> > > How do you expect SimBa to be applied in the discrete control tasks? Will it be enough to beat BBF on Atari as BRO on continuous control?
> > >
> >
> > We appreciate your question regarding the applicability of SimBa to image-based and discrete control tasks. In BBF, a state-of-the-art algorithm for discrete control with image inputs, our simplicity bias philosophy is implicitly incorporated through residual connections and a wider encoder network instead of a deeper one. We believe that adhering to the simplicity bias principle can further enhance performance in the following ways:
> >
> > - **Convolutional Encoder:**
> >   - Normalization Layers: Incorporating normalization layers (e.g., layer-normalization, batch-renormalization) can stabilize training.
> >   - Reduced Non-Linear Activations: Minimizing non-linear activations in the non-linear residual branch aligns with recent practices from advanced vision architectures like ConvNeXtV2 [1].
> > - **Predictor Network:**
> >   - Integrating SimBa: Replacing the MLP with a SimBa network could help to alleviate overfitting present in vision-based training.
> >
> > In summary, by applying the simplicity bias principles to BBF’s architecture, we anticipate further performance improvements on discrete, vision-based control tasks and have included this direction in our future work section.
> >
> > [1] ConvNeXt V2: Co-designing and Scaling ConvNets with Masked Autoencoders., Woo et al.
> >
> >
> > > **Question 3.5**
> > "How do your results about simplicity bias measured through Fourier features relate to this paper [1]? As I understand, authors of [1] claim that neural networks are naturally biased to learn low frequencies faster, and it is hard for them to learn high-frequency signals, which is present in reinforcement learning. SimBa promotes architectures that are biased towards low frequencies. Could you share your perspective on how these findings interact or complement each other?
> >
> >
> > We appreciate the reviewer's insightful question about the relationship between SimBa's simplicity bias and the spectral bias discussed in Yang et al. [1]. While both works address challenges in neural network expressivity for deep RL, they approach the problem from different perspectives.
> >
> > Yang et al. [1] highlight that neural networks naturally prioritize learning low-frequency components over high-frequency ones—a spectral bias that makes it difficult to approximate complex value functions common in RL due to the recursive nature of the Bellman operator. Their work aims to mitigate this spectral bias in underfitted, underparameterized networks, enabling them to learn high-frequency components more effectively.
> >
> > In contrast, SimBa addresses issues arising from overparameterization. While increasing network capacity helps fit complex value functions, it can introduce training instabilities. SimBa leverages a simplicity bias inherent in well-designed overparameterized architectures, promoting the learning of functions that are simple in frequency space at initialization.
> >
> > In essence, while Yang et al [1] enhance underparameterized and underfitted networks to capture high-frequency components, SimBa stabilizes overparameterized and overfitted networks through simplicity bias. Therefore, the choice between these approaches should depend on the specific experimental setup.
> >
> > [1] Overcoming the spectral bias of neural value approximation. Yang et al., ICLR 2022

---

> ### Comment · Reviewer_2Hpz · 2024-11-22
>
> **Regarding the answer to Question 3.1**
> Thank you for the detailed explanation and for conducting the Mann–Whitney U test on SimBa and BRO. I also appreciate the authors' effort to highlight that SimBa’s strengths are not solely limited to achieving high performance but also include computational efficiency and simplicity, as well as its independence from RL-specific techniques such as periodic resets and quantile regression losses.
>
> Therefore, regarding:
>
> > SimBa demonstrates superior scaling performance in terms of average return (...)
>
> I would like to suggest extending the statistical analysis to the architectural comparisons presented in Section 5.2. Specifically, in Figure 5, there is an overlap of standard errors for larger architectures, which raises questions about the statistical significance of SimBa’s reported scaling superiority. Since all architectures are tested in the same algorithmic setup (i.e., without BRO’s RL-specific techniques), it would be valuable to assess the statistical significance of the above claim.
>
> **Regarding the answer to Question 3.2**
> > As shown in Figure 15, the removal of any single component results in a degradation of overall performance, underscoring the necessity of each component for SimBa’s effectiveness. Additionally, we observe that, except for RSNorm, the removal of residual connections, pre-layer normalizations, or post-layer normalizations leads to a reduction in the simplicity score. This further emphasizes the critical role of each individual component in SimBa’s design.
>
> and also in the manuscript one can find:
>
> > Stronger simplicity bias correlates with higher returns for overparameterized networks.
>
> I appreciate the inclusion of Figure 15 and the authors’ effort to quantify the impact of removing individual components. However, I would like to understand better the claim that “stronger simplicity bias correlates with higher returns,” as stated in the manuscript. While this correlation seems evident in Figure 4, it is less apparent in Figure 15. SimBa without RSNorm is one case that questions this correlation.  The second is the SimBa without PostLN also has a higher simplicity score than configurations without residual or preLN components but still worse returns.
>
> Have the authors explicitly measured the correlation between simplicity bias and returns across different configurations? This is particularly important since this relationship forms a central theme of the paper. But, it is also interesting to know if it should be a universal rule of thumb for deep reinforcement learning practitioners. Clarifying this would significantly strengthen the manuscript’s conclusions and provide helpful guidance for the RL community.

---

> ### Author Response · Authors · 2024-11-23
>
> Again, thank you for your constructive review of our manuscript. We appreciate your  feedback and the opportunity to address your concerns. Below, we respond to your specific points:
>
> > 3.1. Statistical Significance of Scaling Performance
>
>
> We appreciate your suggestion to extend the statistical analysis to the architectural comparisons in Section 5.2. In response, we conducted additional Mann–Whitney U tests to evaluate the performance differences between SimBa and BRO across varying critic sizes. The results are summarized below:
>
> | Critic Params (M) | avg. SimBa | avg. BRO | Mann–Whitney U Test (p-value) |
> |-------------------|------------|----------|-------------------------------|
> | 0.07M             | 302.55     | 208.76   | 0.008                         |
> | 0.3M              | 390.68     | 249.47   | 0.008                         |
> | 1.1M              | 594.48     | 419.43   | 0.008                         |
> | 4.5M              | 708.70     | 633.68   | 0.032                         |
> | 17.8M             | 722.42     | 694.39   | 0.310                         |
>
> The analysis reveals that **SimBa significantly outperforms BRO for critic sizes ranging from 0.07M to 4.5M parameters (p < 0.05)**. However, for the largest architecture at 17.8M parameters, the performance difference is not statistically significant (p = 0.310). We speculate that at larger scales, both methods approach similar performance ceilings on the DMC-Hard tasks, thereby reducing the observable differences.
>
> Based on these findings, we acknowledge that the claim of "superior scaling performance" for SimBa requires further nuance. Specifically, while SimBa demonstrates statistically significant scaling advantages for small to medium-sized architectures, its superiority becomes less pronounced for larger architectures. This refined interpretation offers a more balanced view of SimBa’s scaling performance.
>
> Additionally, it is important to note that RSNorm, one of our key contributions, was applied to all experiments to ensure a fair comparison between architectures. Notably, the original BRO architecture without RSNorm exhibits significantly worse performance.
>
> > 3.2. Correlation between Simplicity Bias Score and Performance
>
>
> We recognize the importance of thoroughly investigating the relationship between simplicity bias and performance to substantiate our claims. To address your concerns, we conducted an explicit correlation analysis, detailed in **Appendix D**.
>
> In our study, we computed the correlation between simplicity bias scores and the average returns of 12 distinct architectures, including four primary types: MLP, SimBa, BroNet, and SpectralNet. Additionally, we examined four SimBa variants with component reductions and four SimBa variants with component additions transitioning from MLP to SimBa. As discussed in Appendix C, the simplicity bias measure may not accurately capture the influence of RSNorm. Therefore, we separately evaluated the correlation for architectures with and without RSNorm. All twelve architectures have nearly identical parameter counts, approximately 4.5 million parameters each, with a maximum variation of 1%. Performance was assessed on the DMC-Hard benchmark, trained for one million steps.
>
> The results indicate that architectures employing RSNorm exhibit a **Pearson correlation coefficient of 0.79**, demonstrating a strong positive correlation between simplicity bias score and average return in scaled architectures. In contrast, across all architectures, the **Pearson correlation coefficient is 0.54**, representing a moderate correlation. We attribute the lower correlation in the broader set to the current limitations of the simplicity bias measure.
>
> We believe these findings support our hypothesis that higher simplicity bias scores correlate with improved performance in deep reinforcement learning with scaled-up architectures. We hope this work provides guidance to the RL community in designing effective new architectures.

---

> > ### Author Response · Authors · 2024-11-25
> >
> > Dear Reviewer 2Hpz,
> >
> > Thank you again for your valuable feedback, which has significantly improved the quality of our paper and hypothesis. With two days remaining in the discussion period, we kindly ask if our rebuttal has addressed your concerns.
> > Please let us know if you have further questions or need additional clarification.
> >
> > Best regards,
> > Authors of Submission 3446

---

> > > ### Comment · Reviewer_2Hpz · 2024-11-26
> > >
> > > Thank you for your rebuttal and extra statistics. I will raise my score.

---

### Official Review · Reviewer_F5aT · 2024-11-04

**Soundness:** 4
**Presentation:** 4
**Contribution:** 4
**Rating:** 8
**Confidence:** 3

**Summary:**

The paper introduces SimBa (Simplicity Bias), a novel architecture designed to scale up parameters in deep reinforcement learning (RL) by incorporating simplicity bias. The key components of SimBa are:
- Observation normalization layer
- Residual feedforward block
- Layer normalization

The authors demonstrate that SimBa exhibits higher simplicity bias compared to standard MLPs and shows consistent performance improvements as the number of parameters increases. When integrated with various RL algorithms, including SAC, TD-MPC2, PPO, and METRA, SimBa improves sample efficiency. Moreover, SimBa matches or surpasses state-of-the-art off-policy RL methods across 51 continuous control tasks while maintaining computational efficiency

**Strengths:**

- Novel approach: SimBa addresses an important gap in deep RL research by exploring how to scale up network parameters while leveraging simplicity bias effectively.
- Versatility: The architecture improves sample efficiency across various RL algorithms, including off-policy, on-policy, and unsupervised methods.
- Performance: When applied to SAC, SimBa matches or surpasses state-of-the-art off-policy RL methods across a wide range of tasks.
- Computational efficiency: SimBa achieves high performance without relying on computationally intensive components or complex training protocols.
- Theoretical foundation: The authors provide a clear explanation of simplicity bias and how it's measured, grounding their work in existing theory.
- Significance: This work addresses an important gap in deep RL research by exploring how to effectively scale up network parameters while leveraging simplicity bias. The SimBa architecture offers a promising approach to improve performance across various RL algorithms and tasks without relying on computationally intensive components or complex training protocols

**Weaknesses:**

- While the evaluation covers 51 continuous control tasks, it might be beneficial to see SimBa's performance on a wider range of RL domains, particularly with images.
- The paper doesn't discuss potential limitations or scenarios where SimBa might not be as effective

**Questions:**

- Applicability: Are there any specific types of RL problems or environments where SimBa might not be as effective?
- Why are the standard errors in Figure 10 so high?

---

> ### Author Response · Authors · 2024-11-20
>
> Dear reviewer F5aT,
>
> Thank you for your constructive feedback. To address your concerns, we have provided a detailed response to each comment. In addition, we have provided a common response to address key feedback from all reviewers. Please let us know if you have any further comments or feedback.
>
>
> > **Question 2.1**
> It might be beneficial to see SimBa's performance on a wider range of RL domains, particularly with images.
>
> Thank you for the suggestion to evaluate SimBa on image-based environments. We agree that this is an important direction and have included it in our future work section.
>
> As an preliminary insight, we anticipate that applying SimBa's simplicity bias principles to vision-based control architectures, such as, on top of BBF [1], could provide performance enhancements. Specifically:
>
> - **Convolutional Encoder:**
>   - Normalization Layers: Incorporating normalization layers (e.g., layer-normalization, batch-renormalization) can stabilize training.
>   - Reduced Non-Linear Activations: Minimizing non-linear activations in the non-linear residual branch aligns with practices from advanced vision architectures like ConvNeXtV2 [2].
> - **Predictor Network:**
>   - Integrating SimBa: Replacing the MLP with a SimBa network could help to alleviate overfitting present in vision-based training.
>
> [1] Bigger, Better, Faster: Human-level Atari with human-level efficiency., Schwarzer et al.
>
> [2] ConvNeXt V2: Co-designing and Scaling ConvNets with Masked Autoencoders., Woo et al.
>
>
> > **Question 2.2**
> The paper doesn't discuss potential limitations or scenarios where SimBa might not be as effective
>
> > Are there any specific types of RL problems or environments where SimBa might not be as effective?
> >
>
> While SimBa consistently outperforms or matches naive MLP architectures in over-parameterized regimes across various experiments, its effectiveness may be limited in scenarios with constrained model capacity, such as under-parameterized or on-device settings. In these cases, SimBa's simplicity bias might restrict its ability to effectively fit the target function and could negatively impact parameter count and VRAM usage. Nonetheless, since the majority of the problems we address operate in over-parameterized regimes, we believe SimBa remains highly effective overall.
>
> > **Question 2.3**
> Why are the standard errors in Figure 10 so high?
>
> The high standard errors in Figure 10 are due to the binary nature of task outcomes in Craftax, where each trial results in either success (1) or failure (0).

---

> > ### Comment · Reviewer_F5aT · 2024-11-26
> >
> > Thank you for your detailed responses to my comments and for addressing the concerns raised. Overall, I am satisfied with the revisions and explanations provided. Your responses have strengthened the paper, and I am happy to maintain my positive score.
> > Thank you once again for your thoughtful engagement with my feedback.

---

### Official Review · Reviewer_9P9G · 2024-11-05

**Soundness:** 3
**Presentation:** 3
**Contribution:** 3
**Rating:** 6
**Confidence:** 3

**Summary:**

The paper presents SimBa, an architecture specifically designed for deep reinforcement learning that scales up network parameters by integrating a simplicity bias. The SimBa architecture employs observation normalization, residual feedforward blocks, and layer normalization, fostering simplicity in function representation. By adopting these components, SimBa enhances the sample efficiency and compute efficiency of various RL algorithms. It performs comparably or better than state-of-the-art methods in diverse RL benchmarks, demonstrating effectiveness across off-policy, on-policy, and unsupervised settings.

**Strengths:**

1. **Innovative Application of Simplicity Bias**: The use of simplicity bias in SimBa to manage overparameterization effectively is a novel contribution to deep RL.

2. **Comprehensive Empirical Validation**: SimBa's performance is rigorously tested across multiple RL tasks, including DMC, MyoSuite, and HumanoidBench, showing consistent improvements in efficiency and scalability.

3. **Adaptability**: SimBa’s architecture is algorithm-agnostic and can integrate seamlessly with various RL algorithms, allowing its broad applicability.

4. **Effective Design Choices**: The architectural components (e.g., RSNorm and residual connections) seem thoughtfully selected to reduce complexity while enabling parameter scaling.

**Weaknesses:**

1. Missing relation works about normalization. Researchers show that the RMS Norm works well in training foundation models. The proposed RSNorm is too similar to the RMS Norm, while the author has no discussion about the RMS Norm (even not in related work). Since this work investigates designing and scaling up networks in deep RL, the same tricks in designing and scaling up LLMs should be considered.

2. Unfair comparison between SAC+SimBa v.s. others. Since SAC+SimBa introduces some other layers, it might include more parameters for the neural networks. The author can make some ablations with different sizes of SAC to exclude the capacity issue.

3. Missing details of the model size. I didn't find the detailed sizes of each model. If the sizes are small (such as 10M), the conclusions about the so-called "scaling up" are doubtful.

**Questions:**

1. I am curious about the scaling results on multi-task RL policies, since the proposed architecture is added to the representation part. By the way, maybe the author should also compare the results with baseline methods under scaled-up backbones.

2. The author chose a new metric named simplicity bias score for comparison and analysis. The results also show some effects of the initial parameter distributions. Why not compare with the baseline methods with initialization periodically?

Nikishin, E., Schwarzer, M., D’Oro, P., Bacon, P. L., & Courville, A. (2022, June). The primacy bias in deep reinforcement learning. In International conference on machine learning (pp. 16828-16847). PMLR.

---

> ### Author Response · Authors · 2024-11-20
>
> Dear reviewer 9P9G,
>
> We appreciate your insightful questions and positive support. We have provided a detailed response to each comment. In addition, we have provided a common response to address key feedback from all reviewers.  Please let us know if you have any further comments or feedback.
>
>
> > **Question 1.1**
> The proposed RSNorm is too similar to the RMS Norm, while the author has no discussion about the RMS Norm.
>
> We apologize for causing confusion between **Running Statistic Normalization (RSNorm)** and **Root Mean Square Normalization (RMSNorm)**. Despite the similarity in their names, RSNorm and RMSNorm differ fundamentally in two key aspects:
>
> 1. **Normalization Axis:**
>    - **RSNorm** normalizes each feature dimension independently across the batch, similar to Batch Normalization. For example, if one feature is sampled from N(0,1) and another from N(10,1), RSNorm scales each to N(0,1), preventing any single feature from dominating due to significant differences in scale.
>    - **RMSNorm** normalizes features within each sample, which may result in features with larger variances to disproportionately influence the model.
> 2. **Statistics Used:**
>    - **RSNorm** utilizes cumulative running statistics for normalization. This approach is more stable and less noisy compared to using mini-batch statistics, which is particularly beneficial in reinforcement learning settings where batch estimates can be unreliable.
>    - **RMSNorm** does not rely on any mini-batch based statistics, so it doesn't need to track running statistics.
>
>
> To further address your concern, we have now included a discussion of RMSNorm in our manuscript and conducted comparative experiments on DMC-Hard averaged over 5 random seeds. The results are summarized below:
>
> | Input Normalization   | Mean Score |
> |-----------------------|------------|
> | None                  | 353        |
> | LayerNorm             | 354        |
> | RMSNorm               | 402        |
> | RSNorm.               | 710        |
>
> As shown, RSNorm significantly outperforms RMSNorm and other normalization methods, highlighting the effectiveness of RSNorm. We have also revised Figure 12 to include RMSNorm as a baseline for clearer comparison.
>
>
>
>
> > **Question 1.2**
> Unfair comparison between SAC+SimBa vs others. Since SAC+SimBa introduces some other layers, it might include more parameters for the neural networks. The author can make some ablations with different sizes of SAC to exclude the capacity issue.
>
> >Missing details of the model size. I didn't find the detailed sizes of each model.
>
> We appreciate your concern regarding potential capacity differences between SimBa and other architectures (MLP, BRONet, SpectralNet). To ensure a fair comparison, we have provided detailed parameter counts with a hidden-dimension $d$ and num-blocks $2$.
>
> - MLP: $17d^2$.
> - SpectralNet: $17d^2 + 8d$.
> - BroNet: $17d^2 + 10d$.
> - SimBa: $17d^2 + 8d$.
>
> Additionally, we present the proportion of parameters contributed by normalization layers in SimBa:
>
>    | Hidden Dimension | Total Parameters | Normalization Layers (%) |
>    |------------|-------------------|--------------------------|
>    | 64         | 0.07M             | 0.7%                     |
>    | 128        | 0.3M              | 0.36%                    |
>    | 256        | 1.1M              | 0.18%                    |
>    | 512        | 4.5M              | 0.09%                    |
>    | 1024       | 17.8M             | 0.05%                    |
>
> As shown, normalization layers contribute less than 1% of the total parameters across all model sizes, making their impact on model capacity negligible. Therefore, the superior performance of SAC+SimBa stems from the effective integration of normalization layers rather than an increased model capacity. To improve clarity, we have updated Figure 5 to display the number of parameters on the x-axis instead of the hidden dimension, accurately reflecting our comparisons.

---

> > ### Author Response · Authors · 2024-11-20
> >
> > > **Question 1.3**
> > If the sizes are small (such as 10M), the conclusions about the so-called "scaling up" are doubtful.
> >
> > In deep RL, commonly adopted baseline network architectures typically consist of 3 to 5 linear layers, resulting in architectures with approximately 0.1 million parameters. Therefore, a model size of 10 million parameters is considered substantial in the deep RL literature, as it is 100 times larger than commonly adopted architectures. Historically, RL struggles with scaling due to inefficiencies and limited performance gains [1,2]. Compared to recent works, our scaling approach is competitive:
> >
> > - DDPG [3], SAC[4]: $\approx 0.1M$.
> > - TD7 [5]: $\approx 1M$.
> > - REDQ [6]: $\approx 2M$.
> > - TDMPC2 [7]: $\approx 5M$.
> > - SimBa: Up to $17.8M$.
> >
> > Our largest model, with 17.8M parameters, aligns with recent scaling efforts in RL [8,9], which scale up to approximately 20M parameters. We observed that scaling beyond 5M parameters yields diminishing returns, likely because controlling agents with low-dimensional, state-based inputs ($\approx$200 dimensions) is less complex than tasks in NLP or computer vision, which involve high-dimensional inputs ($\approx$100,000 dimensions).
> >
> > Future work could explore scaling SimBa for vision-based tasks, further evaluating the scalability of our SimBa which we had included it in our future work section.
> >
> > [1] Training Larger Networks for Deep Reinforcement Learning, Ota et al, arXiv 2021.
> >
> > [2] Towards Deeper Deep Reinforcement Learning with Spectral Normalization, Bjorck et al, NeurIPS 2021.
> >
> > [3] Continuous control with deep reinforcement learning, Lilicrap et al, ICLR 2016.
> >
> > [4] Soft Actor-Critic: Off-Policy Maximum Entropy Deep Reinforcement Learning with a Stochastic Actor, Haarnoja et al, ICML 2018.
> >
> > [5] For SALE: State-Action Representation Learning for Deep Reinforcement Learning., Fujimoto et al., NeurIPS 2023.
> >
> > [6] Randomized Ensembled Double Q-Learning: Learning Fast Without a Model., Chen et al., ICLR 2021.
> >
> > [7] TD-MPC2: Scalable, Robust World Models for Continuous Control., Hansen et al., ICLR 2024.
> >
> > [8] Bigger, Better, Faster: Human-level Atari with human-level efficiency., Schwarzer et al., ICML 2023.
> >
> > [9] Bigger, Regularized, Optimistic: scaling for compute and sample-efficient continuous control., Nauman et al., NeurIPS 2024.
> >
> >
> >
> >
> > > **Question 1.4**
> > I am curious about the scaling results on multi-task RL policies, since the proposed architecture is added to the representation part.
> >
> > We appreciate the reviewer’s interest in evaluating SimBa on multi-task RL policies. Multi-task RL, by its own, presents significant challenges, such as representation conflicts [1] and differences in reward scaling [2]. In addition, standard multi-task RL benchmarks, like Meta-World [3], often do not scale effectively with increased model complexity.
> >
> > Integrating SimBa into multi-task settings requires advanced strategies beyond our current scope. However, we recognize its potential and have included it as a direction for future work in the revised manuscript.
> >
> > [1] Distral: Robust Multitask Reinforcement Learning, Teh et al, NeurIPS 2017.
> >
> > [2] Multi-task Deep Reinforcement Learning with PopArt, Hessel et al, AAAI 2019.
> >
> > [3] Meta-World: A Benchmark and Evaluation for Multi-Task and Meta Reinforcement Learning, Yu et al, CoRL 2019.
> >
> >
> > > **Question 1.5**
> > The author chose a new metric named simplicity bias score for comparison and analysis. The results also show some effects of the initial parameter distributions. Why not compare with the baseline methods with a periodic reinitialization?
> >
> > Thank you for the suggestion. However, measuring simplicity bias score with periodic reinitialization presents challenges:
> >
> > - Using a uniform input distribution results in the same simplicity bias scores after reinitialization.
> > - Measuring simplicity bias during training or with repeated initialization introduces significant variability. As noted on page 3, "directly comparing simplicity bias across different architectures after convergence is challenging due to the randomness of the non-stationary optimization process, especially in RL, where the data distribution changes continuously."
> >
> > Therefore, we focus on evaluating simplicity bias score at initialization to ensure stable and meaningful comparisons across architectures.

---

> > > ### Author Response · Authors · 2024-11-25
> > >
> > > Dear Reviewer 9P9G,
> > >
> > > Thank you again for taking the time to review our paper. With two days remaining in the discussion period, we kindly ask you to consider our rebuttal and let us know if you have any further concerns or questions.
> > >
> > > Best regards,
> > > Authors of Submission 3446

---

### Author Response · Authors · 2024-11-20

### General Response

We sincerely appreciate the constructive and insightful feedback from all reviewers.

The key requests and response for the reviews are as follows:

**1. Extending Training of SimBa from 1M to 5M Steps**

   We have extended the training of SAC with SimBa on DMC-Hard to 5 million steps, comparing it with TD-MPC2 and SAC with MLP. The results demonstrate that SimBa remains stable and robust during extended training, exhibiting a better learning curve than the baseline methods. Detailed results are provided in **Appendix C.1.**

**2. Ablation Study of Excluding Each Component of SimBa**

   We conducted an ablation study by excluding each component of SimBa individually. The findings indicate that each component contributes to improved performance, validating the efficacy of our design choices. This study is detailed in **Appendix C.2.**

**3.Detailed Plasticity Analysis Including DMC Easy & Medium**

   We performed a plasticity analysis on DMC Easy and Medium tasks, comparing SimBa with the MLP architecture. Even in simpler environments, SimBa showed a lower dormant ratio and feature norm, indicating reduced plasticity loss. These results are included in **Appendix D.**

**4.Comparison of RSNorm to RMSNorm**

   We added RMSNorm (Root Mean Square Normalization) as a baseline for observation normalization. Our experiments reveal that RSNorm significantly outperforms RMSNorm. The comparative results are presented in **Figure 12.**

**5.Refining Future Work**

   Based on the reviewers' feedback, we have revised the **future work section** to discuss the extension of SimBa to vision-based and multi-task reinforcement learning.

In conclusion, we hope our revisions address the concerns raised by reviewers. We look forward to any further feedback.

Warm regards,

Authors of Submission 3446

---

### Meta-Review · Area_Chair_o2X4 · 2024-12-14

**Metareview:**

This paper introduces a simple and effective architecture(SimBa) to scale up network parameters in a deep RL setting, by defining and leveraging a simplicity bias score. SimBa consists of three key components that are drop-in replacement to existing RL methods: RSNorm (running statistics norm), residual feedforward blocks and layer norm. The key idea is that by promoting simplicity bias SimBa enhances performance in over-parametrized regimes. SimBa is shown to improve sample & computational efficiency, and overall performance across a variety of standard RL environments and algorithms. The key finding then is that by using a simplicity bias, we can effectively scale parameters without algorithmic modifications, achieving competitive or superior results compared to SOTA methods.

I think this paper proposes a method that is simple and effective, with clear strengths. Leveraging simplicity bias is a relatively under explored topic in RL. It likely has relationship to algorithmic kolmogorov complexity and should be studied further to discover simple architectural changes that could yield better generalization in neural networks. The experimental results are rigorous, spanning more than 50 tasks. The simplicity bias score is theoretically motivated and makes sense to me.

The biggest limitation of SimBa is that its only evaluated on state space RL tasks - there is no vision or large observation spaces. This will limit its applicability but I do not think this is a severe limitation. I think it is important and interesting to study these problems in isolation. We will learn something new when the authors or other researchers do further studies in that direction and compare it back to this paper. Another problem with the simplicity score framework is to discuss and highlights failure modes - what happens in under parametrized regimes or what are other failure modes?

Despite these limitations, I think the paper is strong enough and there is clear consensus amongst the authors regarding its strength. I would recommend accepting this paper and it will be fruitful for the ICLR community to explore more in this direction.

**Additional Comments On Reviewer Discussion:**

During the discussion period, reviewers raised several concerns but the major ones seem to be resolved. The biggest one was raised by 2Hpz and others who highlighted the lack of statistical significance in the results. The authors acknowledged and conducted Mann-Whitney U tests across environments and architectures, showing statistical significant performance improvements.

Reviewers 9P9G and 2Hpz requested more detailed ablation studies to clarify the contributions of SimBa’s components. The authors added additional analyses and plasticity metrics, demonstrating how each component contributed to improved performance and simplicity bias. Reviewer 8rZo called for longer training experiments, prompting the authors to extend evaluations to 5 million steps on DMC-Hard tasks. Provided that these improvements are included in the final version of the paper, I believe the authors have addressed the major concerns.

---

### Decision · Program_Chairs · 2025-01-22

Accept (Spotlight)